# Multivariate analysis of *Iris meda* Stapf based on phenological and morphological characteristics

**Alireza Khaleghi[1]\*, Ali Khadivi**![ORCID]**[1]\*, Yazgan Tunç[2]\***

**1** Department of Horticultural Sciences, Faculty of Agriculture and Natural Resources, Arak University, Arak, Iran, **2** Republic of Türkiye, Ministry of Agriculture and Forestry, General Directorate of Agricultural Research and Policies, Hatay Olive Research Institute Directorate, Hassa, Hatay, Türkiye

\* a-khaleghi@araku.ac.ir (AK); a-khadivi@araku.ac.ir (AK); yazgan.tunc@tarimorman.gov.tr (YT)

## Abstract

The genus *Iris* represents one of the most diverse and horticulturally valuable ornamental groups worldwide, with considerable ecological, morphological, and genetic importance. However, despite its significance, many species remain poorly characterized, and the extent of intraspecific variability is largely underexplored, limiting their effective utilization in breeding and conservation. *Iris meda* Stapf, a narrowly distributed yet morphologically diverse species native to the Irano-Turanian region, was investigated in this study to address this knowledge gap and to assess its phenotypic variation. A total of 108 wild accessions from the Tafresh region of Iran were evaluated based on 41 morphological traits. Comprehensive multivariate analysis revealed substantial phenotypic diversity, with particularly high coefficients of variation in leaf shape (62.56%), crest color (59.79%), and standard color (46.35%). Elite genotypes such as 'I. meda-3', 'I. meda-7', and 'I. meda-36' were identified for their superior floral traits, while morphologically divergent outliers—including 'I. meda-10', 'I. meda-23', and 'I. meda-59'—exhibited notable divergence in PCA biplots. Strong correlations were observed between flower diameter and surface area ($r = 0.80$; $\beta = 0.87$) and between fall length and beard length ($\beta = 0.63$). Principal component analysis indicated that floral architecture, vegetative growth, and pigmentation were the primary contributors to variation. These findings highlight the considerable genetic resources available in *I. meda* for ornamental breeding and conservation programs.

## 1. Introduction

The genus *Iris* L., belonging to the family Iridaceae, constitutes a taxonomically intricate and morphologically diverse group comprising more than 300 recognized species, primarily distributed across temperate regions of the Northern Hemisphere [1]. These species exhibit remarkable ecological plasticity, adapting to a wide range of environments from arid steppes to alpine meadows [2]. This adaptability is coupled with substantial morphological diversification, particularly in floral traits, which

**Data availability statement:** All relevant data are within the paper and its Supporting information files.

**Funding:** The author(s) received no specific funding for this work.

**Competing interests:** The authors have declared that no competing interests exist.

has rendered *Iris* a diverse ornamental species of global importance and one of the most attractive genera for both horticultural cultivation and scientific investigation [3,4]. The genus is widely appreciated for its striking floral aesthetics, characterized by elaborate structures, vibrant pigmentation, and species-specific morphological patterns, making it a popular subject of breeding programs aimed at developing novel ornamental cultivars [5]. Indeed, Iris has been cultivated for centuries, reflecting its enduring horticultural and cultural value.

Among the species within this genus, *Iris meda* Stapf represents a particularly intriguing taxon due to its limited geographic distribution and high degree of morphological variability [6]. Native to the Irano-Turanian phytogeographical region, *I. meda* thrives in montane and subalpine habitats across central and western Iran, often occupying ecologically marginal zones where environmental conditions exert strong selective pressures. Despite its restricted range, *I. meda* exhibits a wide array of phenotypic expressions in both vegetative and reproductive organs, suggesting a significant degree of genetic diversity shaped by microhabitat differentiation and possibly historical gene flow. This diversity, however, remains underexplored, as the species has received relatively limited attention in systematic, ecological, and breeding-oriented studies.

Assessing genetic diversity is fundamental for plant breeding and conservation programs [7]. In addition to traditional morphological characterization, various molecular methods are widely employed. Randomly amplified polymorphic DNA (RAPD), simple sequence repeats (SSR/microsatellites), single nucleotide polymorphisms (SNP), and next-generation sequencing (NGS)-based approaches that provide high-resolution insights into genetic variation have been effectively applied in the analysis of plant genetic resources [8]. Recent studies have demonstrated that combining morphological and molecular data produces more reliable outcomes in understanding population structure and detecting patterns of environmental adaptation [9]. In the case of *Iris* species, these approaches have been successfully utilized to elucidate both interspecific and intraspecific variation, with SSR and ISSR markers in particular being reported as highly informative for detecting genetic diversity [10,11]. Such findings highlight the critical importance of integrated diversity analyses not only for taxonomic clarification but also for identifying valuable genetic resources for ornamental breeding.

The taxonomic complexity of *Iris* species is largely attributed to frequent hybridization events, polyploidy, and high phenotypic plasticity, which complicate species delimitation and intraspecific classification [12]. In such contexts, morphological characterization serves as a vital tool not only for understanding species boundaries but also for identifying populations with desirable traits for conservation and horticultural applications [13]. Previous studies on other *Iris* species, including *I. germanica*, *I. pseudacorus*, *I. maackii, I. japonica, I. hymenospatha,* and *I. histrio* have demonstrated that multivariate analyses based on a comprehensive set of morphological descriptors—particularly floral traits such as flower diameter, petal shape, and pigmentation—can effectively discriminate among accessions and reveal underlying genetic structures within natural populations [13–17].

Despite these advances, *I. meda* has received limited attention in both morphological and molecular research. This study represents the most comprehensive morphological assessment of *I. meda* to date, involving 108 wild accessions evaluated across 41 traits. We hypothesize that although the species is geographically restricted, it harbors substantial phenotypic and genetic diversity shaped by local adaptation. We further propose that this diversity constitutes a valuable genetic resource for germplasm conservation and ornamental breeding.

Given the increasing importance of conserving endemic plant taxa and the growing interest in utilizing native species for ornamental horticulture, a thorough understanding of the morphological variability of *I. meda* is both timely and necessary. Floral traits, which play a crucial role in pollinator attraction and reproductive success, are of particular relevance for ecological and evolutionary studies [18]. Additionally, vegetative characters such as leaf morphology and stem architecture provide important insights into environmental adaptation and plant vigor, further enhancing the value of integrated morphological assessments [17].

The present study addresses this gap by conducting a detailed morphological evaluation of 108 wild accessions of *I. meda* collected from the Tafresh region of Markazi Province in Iran, a region characterized by its unique climatic and topographical features. A total of 41 morphological traits, encompassing both quantitative and qualitative characters related to leaf, stem, and floral structures, were examined to capture the full spectrum of phenotypic diversity within the species. The research employed a suite of statistical techniques, including analysis of variance, Pearson's correlation, principal component analysis, hierarchical cluster analysis, and multiple regression analysis, to interpret the complex interrelationships among traits and to identify key characters contributing to intraspecific variation.

By integrating detailed morphological data with robust multivariate statistical methods, this study aims to (i) quantify the extent of phenotypic variation present within *I. meda*, (ii) identify accessions with distinctive or superior traits that may be useful for ornamental breeding and conservation strategies, and (iii) elucidate the underlying structure of morphological diversity within the species. The findings are expected to enhance the current understanding of species-level variation in *Iris*, provide a valuable reference for future taxonomic and ecological research, and support the development of targeted conservation programs aimed at preserving the genetic integrity and adaptive potential of native *I. meda* populations.

## 2. Materials and methods

### 2.1. Plant material

In May 2023, the morphological variation of 108 wild accessions of *Iris meda* was evaluated in natural habitats of the Tafresh region, Markazi Province, Iran. All sampled plants were mature, naturally growing individuals collected from wild populations. To minimize the possibility of sampling clonal individuals, a minimum distance of 200 m was maintained between specimens. The sampling sites were geographically located between 34°37′25″ and 34°39′23″ N latitude and 50°04′10″ and 50°05′17″ E longitude, at elevations ranging from 2,227–2,600 m above sea level (Fig 1). The map in Fig 1 was generated using ArcGIS software version 10.1 [19]. The regional climate is characterized by annual temperature extremes from −20.5 °C to +39 °C and a long-term mean annual precipitation of 314 mm.

The climate of Tafresh is semi-arid according to the de Martonne climate classification and semi-arid and cold according to the Amberger climate classification, and it is mostly covered in snow during the winters. In the year 2023, the annual minimum, mean, and maximum temperatures of the Tafresh area were 7.4, 13.3, and 19.2 °C, respectively; and the annual relative humidity was 44.00%. *I. meda* stands out as one of the most significant plant types and species in the Tafresh area.

The formal identification of the specimens was performed by Dr. Alireza Khaleghi. A herbarium voucher specimen with sediment number IM-4435 has been donated to the public available herbarium of the Faculty of Agriculture and Natural Resources of Arak University, Iran.

**2.1.1. Statement specifying permissions.** For this study, we acquired permission to study *I. meda* issued by the Agricultural and Natural Resources Ministry of Iran.

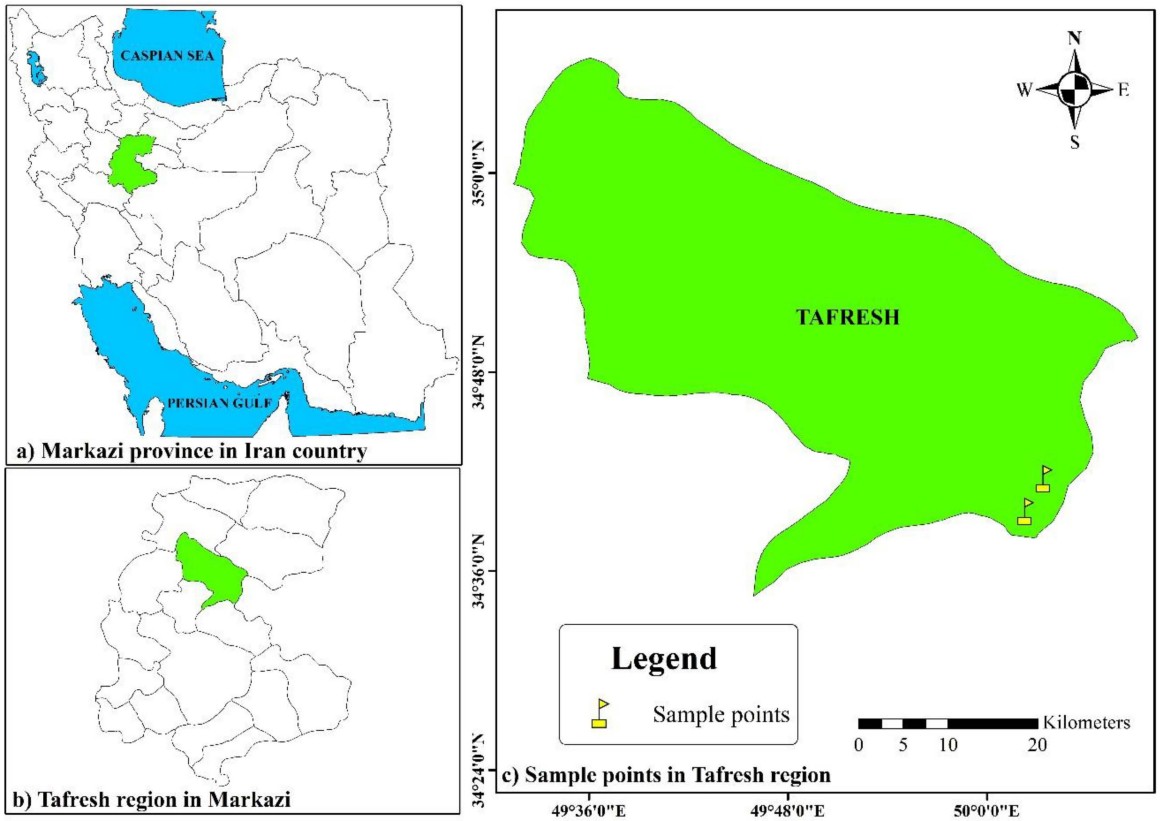

**Fig 1. Geographic locations of collection sites of the studied *Iris meda* accessions.** The map was generated by the second author, Ali Khadivi.

**2.1.2. Statement on experimental research and field studies on plants.** The either cultivated or wild-growing plants sampled comply with relevant institutional, national, and international guidelines and domestic legislation of Iran.

## 2.2. Morphological evaluations

A total of 41 morphological traits associated with stem, leaf, and floral structures were evaluated. All morphological measurements were conducted at the time of flowering to ensure consistency across observations. Quantitative traits—including plant height, stem height, stem thickness, peduncle thickness, peduncle length, and the width and length of various floral organs such as the spathe, flower, fall, crest, and standard, along with anther and filament lengths—were measured using a precision digital caliper (Mitutoyo® Absolute Digimatic Caliper, 0–150 mm range, ± 0.01 mm accuracy). The extent of the signal patch over the fall was expressed as the ratio of patch surface area to fall width, while the flower shape was represented by the ratio of flower diameter to flower height, as described by Sapir et al. [20]. Furthermore, flower surface area and patch surface area were estimated using specific formulas outlined by Sapir et al. [20] (Equations 1 and 2).

$$Flower\ surface\ =\ flower\ diameter\ \times\ flower\ height\ (in\ cm^2) \tag{1}$$

$$Patch\ surface\ =\ patch\ length\ \times\ width\ (in\ cm^2) \tag{2}$$

Qualitative characteristics—including leaf shape and color, flower scent, and the coloration of the fall, standard, and crest—were assessed through standardized coding systems, categorical scoring methods, and structured rating scales to ensure consistent and reproducible evaluations.

## 2.3. Statistical analysis

In order to evaluate the variability among the studied accessions based on the measured traits, a one-way analysis of variance (ANOVA) was performed at a 5% probability threshold ($p < 0.05$) using JMP® Pro 17 software [21]. The interrelationships among these traits were investigated through the computation of Pearson's correlation coefficients (r), employing Origin Pro® 2025b software [22]. Principal component analysis was also conducted using the same software to identify the key variables contributing to genotypic divergence. To enhance the interpretability of the principal components, Varimax rotation with Kaiser Normalization was applied, enabling clearer delineation of trait-component loadings. For classification purposes, hierarchical cluster analysis was carried out based on Ward's linkage method and Euclidean distance, as implemented in Origin Pro® 2025b. Before clustering, data preprocessing involved min-max normalization (scaling between 0 and 1) followed by Z-score standardization, ensuring comparability across traits and minimizing scale-related biases [23]. A biplot was generated using the first two principal components (PC1 and PC2) to visualize the spatial distribution of both accessions and traits in a reduced dimensional space. In addition, PC scores were calculated for each genotype based on the PCA data set obtained from the *I. meda* accessions. This analysis was conducted to summarize the multidimensional variation among genotypes and to reveal the structural differentiation among the accessions. Furthermore, multiple linear regression analysis was applied to determine which traits significantly influenced floral characteristics, designated as dependent variables. This analysis was executed via the stepwise selection approach within SPSS® software (SPSS Inc., Chicago, IL, USA), in line with the methodologies outlined by Norusis [24], Efe et al. [25], and SPSS [26].

## 3. Results and discussion

### 3.1. Descriptive statistics among accessions

Descriptive data on morphological traits observed in the examined *I. meda* accessions are summarized in Table 1. Results from the one-way ANOVA ($p < 0.05$) revealed statistically significant differences among the accessions. The highest phenotypic variation was recorded for leaf shape (62.56%), crest color (59.79%), style flap color (58.37%), standard color (46.35%), and patch color (40.37%). In contrast, the lowest variability was observed in traits such as standard length (10.08%), fall length (9.48%), style arm length (9.26%), crest length (9.26%), and crest width (8.76%). Importantly, 18 out of the 41 assessed traits (equating to 43.90%) exhibited coefficients of variation (CV) above 20.00%, indicating a high level of genetic diversity among the studied accessions [27].

Traits with CV values exceeding 20% are considered highly variable and are thus regarded as reliable indicators for distinguishing between accessions, genotypes, or cultivars [28]. Furthermore, traits with broader quantitative ranges often align with higher CV values, reinforcing their applicability in breeding program selection strategies [29]. Conversely, traits characterized by lower CVs suggest more consistent expression across genotypes, indicating greater phenotypic stability (Khadivi-Khub and Etemadi-Khah, 2015). Similar patterns have been reported in other fruit crops; for instance, pomegranate has shown a CV as high as 47.82%, emphasizing significant phenotypic diversity [30].

Plant length exhibited considerable variation, ranging from 11.30 cm in accession '*I. meda*-9' to 21.40 cm in '*I. meda*-15', which suggests a strong genotypic effect on vegetative growth performance. A comparable trend was observed for stem length, with values extending from 20.97 mm in '*I. meda*-9' to 104.50 mm in '*I. meda*-15', indicating that specific accessions may possess enhanced potential for shoot elongation. The concurrent highest values recorded in '*I. meda*-15' for both traits implies a positive association between overall plant height and stem development. In terms of floral characteristics, a wide range of variation was recorded in spathe length, from 42.55 mm in '*I. meda*-35' to 71.88 mm in '*I. meda*-3'. Similarly, flower diameter and flower length demonstrated substantial inter-accession differences, with diameter

**Table 1. Summary of descriptive statistical parameters for morphological traits evaluated in *Iris meda* accessions.**

| Trait | Abb | Unit | Min | Max | Mean | ±SD | CV (%) |
|---|---|---|---|---|---|---|---|
| Plant length | V1 | cm | 11.30 | 21.40 | 16.14* | ±2.34 | 14.47 |
| Stem length | V2 | mm | 20.97 | 104.50 | 60.50* | ±18.07 | 29.87 |
| Stem diameter | V3 | mm | 2.48 | 4.29 | 3.42* | ±0.45 | 13.12 |
| Peduncle diameter | V4 | mm | 2.67 | 4.86 | 3.59* | ±0.50 | 13.83 |
| Leaf number | V5 | Number | 3 | 6 | 4.48* | ±0.65 | 14.46 |
| Bottommost leaf length | V6 | cm | 7.80 | 16.40 | 12.31* | ±1.88 | 15.26 |
| Bottommost leaf width | V7 | mm | 6.27 | 18.10 | 11.63* | ±2.87 | 24.66 |
| Leaf shape | V8 | Code | 1 | 5 | 2.07* | ±1.30 | 62.56 |
| Leaf color | V9 | Code | 1 | 5 | 2.96* | ±1.09 | 36.93 |
| Spathe length | V10 | mm | 42.55 | 71.88 | 57.52* | ±6.27 | 10.91 |
| Spathe width | V11 | mm | 14.00 | 28.65 | 20.44* | ±3.42 | 16.73 |
| Flower diameter | V12 | mm | 41.56 | 71.23 | 52.87* | ±5.98 | 11.30 |
| Flower length | V13 | mm | 44.68 | 67.84 | 53.57* | ±5.67 | 10.58 |
| Flower diameter/length | V14 | Ratio | 0.71 | 1.37 | 0.99* | ±0.13 | 13.26 |
| Flower surface | V15 | cm² | 19.62 | 42.96 | 28.40* | ±4.94 | 17.38 |
| Flower scent | V16 | Code | 2 | 6 | 4.11* | ±1.09 | 26.47 |
| Fall length | V17 | mm | 37.80 | 54.63 | 46.54* | ±4.41 | 9.48 |
| Fall width | V18 | mm | 15.67 | 26.47 | 20.60* | ±2.90 | 14.07 |
| Fall color | V19 | Code | 1 | 9 | 6.11* | ±2.03 | 33.29 |
| Standard length | V20 | mm | 46.30 | 68.42 | 55.53* | ±5.60 | 10.08 |
| Standard width | V21 | mm | 15.90 | 30.16 | 22.68* | ±3.03 | 13.37 |
| Standard color | V22 | Code | 1 | 10 | 5.39* | ±2.50 | 46.35 |
| Crest length | V23 | mm | 26.67 | 38.80 | 33.98* | ±3.15 | 9.26 |
| Style arm length | V24 | mm | 21.90 | 31.36 | 26.48* | ±2.45 | 9.26 |
| Style crest length | V25 | mm | 3.78 | 10.32 | 7.49* | ±1.32 | 17.57 |
| Crest width | V26 | mm | 13.45 | 19.41 | 16.53* | ±1.45 | 8.76 |
| Crest color | V27 | Code | 1 | 9 | 3.83* | ±2.29 | 59.79 |
| Beard length | V28 | mm | 19.62 | 33.15 | 26.20* | ±3.08 | 11.75 |
| Beard width | V29 | mm | 2.90 | 8.01 | 5.66* | ±1.43 | 25.18 |
| Beard color | V30 | Code | 1 | 7 | 4.72* | ±1.58 | 33.52 |
| Patch width | V31 | mm | 5.76 | 14.78 | 10.47* | ±2.59 | 24.73 |
| Patch length | V32 | mm | 6.35 | 15.71 | 11.20* | ±2.55 | 22.75 |
| Patch surface | V33 | cm² | 0.37 | 2.08 | 1.22* | ±0.49 | 39.91 |
| Patch surface/fall width | V34 | Ratio | 0.19 | 0.99 | 0.58* | ±0.20 | 34.73 |
| Patch color | V35 | Code | 1 | 9 | 4.56* | ±1.84 | 40.37 |
| Style flap length | V36 | mm | 1.10 | 2.60 | 1.79* | ±0.32 | 18.07 |
| Style flap color | V37 | Code | 1 | 8 | 3.19* | ±1.86 | 58.37 |
| Anther length | V38 | mm | 12.26 | 19.87 | 15.10* | ±1.60 | 10.61 |
| Anther color | V39 | Code | 1 | 7 | 4.94* | ±1.86 | 37.73 |
| Filament length | V40 | mm | 7.77 | 13.62 | 10.24* | ±1.35 | 13.22 |
| Filament color | V41 | Code | 1 | 7 | 4.61* | ±1.24 | 26.83 |

*Abb* Abbreviations, *Max* Maximum, *Min* Minimum, *±SD* Standard Deviation, *CV* Coefficient of Variation. *One-way analysis of variance (ANOVA) for the measured characteristics is significant at the 5% level ($p < 0.05$).

ranging from 41.56 mm in 'I. meda-85' to 71.23 mm in 'I. meda-36', and length from 44.68 mm in 'I. meda-23' to 67.84 mm in 'I. meda-2'. The largest floral dimensions observed in 'I. meda-36' and 'I. meda-2' underscore their potential ornamental value and aesthetic appeal. Petal-related traits—fall length (37.80 mm in 'I. meda-21' to 54.63 mm in 'I. meda-7'), fall width (15.67 mm in 'I. meda-21' to 26.47 mm in 'I. meda-1'), and standard petal length (46.30 mm in 'I. meda-23' to 68.42 mm in 'I. meda-2')—also revealed pronounced phenotypic variability. Accessions such as 'I. meda-7', 'I. meda-1', and 'I. meda-2' exhibited superior petal dimensions, which may confer an advantage in floral display and reproductive efficacy. These particular traits, especially the dimensions of the fall and standard petals, are likely to influence pollinator attraction and should therefore be prioritized in future studies on reproductive ecology and ornamental breeding. Finally, style crest length showed notable variation, ranging from 3.78 mm in 'I. meda-19' to 10.32 mm in 'I. meda-56'.

The considerable variability observed across both vegetative and reproductive characteristics indicates a substantial degree of genetic diversity within the examined Iris population. This morphological differentiation provides a valuable resource for potential selection and breeding efforts. Particularly, the floral traits with broad variation hold significant promise for ornamental enhancement. The extent of phenotypic divergence underscores the suitability of I. meda as a candidate for both conservation initiatives and horticultural exploitation.

Similar trends of morphological variation have been consistently reported in prior research on Iris taxa. For instance, Azimi et al. [31] highlighted notable diversity in plant stature and floral attributes among Iranian Iris species, underlining their potential for taxonomic clarification and germplasm utilization. In a related study, Azimi et al. [32] applied multivariate statistical techniques to I. germanica hybrids, identifying key floral characteristics that effectively discriminated among genotypes and were beneficial for hybrid selection. Likewise, Asgari et al. [33] documented extensive morphological variation in wild Iris species with ornamental value, stressing the importance of floral dimensions in determining aesthetic appeal and commercial potential. Ghorbani et al. [34] also reported significant morphological diversity in I. pseudacorus accessions, especially regarding vegetative vigor and reproductive organs, suggesting the existence of regionally adapted forms with conservation importance.

Together, these findings corroborate the present study's conclusions and emphasize that both vegetative and reproductive morphological traits are essential indicators of genetic diversity and breeding potential within the genus Iris. The recurrence of such patterns across different Iris species and environmental conditions supports the reliability and relevance of morphological assessments as a foundational approach for both genetic improvement and conservation planning.

Comparable levels of morphological and biochemical variability have also been reported in other cucurbit and underutilized species such as Coccinia and Momordica dioica [35,36].

The frequency distribution data of qualitative morphological traits among the I. meda accessions, as presented in Table 2, provides crucial insights into the intra-specific diversity and phenotypic plasticity of the studied population. A careful evaluation of this data reveals several noteworthy trends and implications relevant to the taxonomy, ecology, and potential selective breeding of this species.

The diversity observed in qualitative morphological characters can be interpreted as follows:

The variation in leaf shape is relatively constrained, with three dominant forms observed: elongated (59 accessions), slightly curved (40), and curved (9). The predominance of elongated leaves suggests a possible stabilizing selection for this trait, which may be ecologically advantageous under the prevailing habitat conditions. Conversely, the presence of curved and slightly curved forms may indicate ongoing differentiation or plastic responses to microhabitat variability.

Leaf color exhibits moderate variation, with green (76 accessions) being the most frequent phenotype, followed by light green (17) and dark green (15). This gradient may reflect differences in chlorophyll content or light adaptation strategies, potentially influenced by soil type, canopy cover, or elevation.

Flower scent intensity spans a wide spectrum, ranging from very little (6 accessions) to very high (6), with the highest frequencies recorded for high (46) and little (34) scent. This continuous variation points to underlying genetic diversity and suggests a potential role in pollinator interactions. The presence of multiple scent categories might indicate adaptation to a diverse pollinator assemblage or selective pressures associated with pollination syndromes.

Table 2. Frequency distribution of qualitative morphological features observed among the assessed *Iris meda* accessions.

| Trait | Frequency (No. of accessions) | | |
| --- | --- | --- | --- |
| | 1 | 2 | 3 |
| Leaf shape | Elongated (59) | – | Slightly Curved (40) |
| Leaf color | Light green (17) | – | Green (76) |
| Flower scent | – | Very little (6) | Little (34) |
| Fall color | Milky background + Pale brown veins + No spot on the edge (3) | Milky background + Brown veins + No spot on the edge (3) | Milky background + Brown veins + Limited spot on the edge (12) |
| Standard color | Milky background + Pale gray edge (9) | – | Milky background + Gray/brown edge (21) |
| Crest color* | S.A.: Pale yellow/Pale brown + S.C.: Yellow/Pale brown (27) | S.A.: Pale yellow/brown + S.C.: Yellow/Pale brown (15) | S.A.: Milky/brown + S.C.: Milky/Pale brown (3) |
| Beard color | Milky (6) | – | Pale yellow (24) |
| Patch color | Very pale purple (3) | – | Pale purple (45) |
| Stigmatic lip color | Cream (6) | Cream/yellow (57) | Cream/Pale brown (12) |
| Anther color | White/Milky (6) | – | Milky (30) |
| Filament color | White (3) | – | White/Milky (24) |
| Trait | Frequency (No. of accessions) | | |
| | 4 | 5 | 6 |
| Leaf shape | – | Curved (9) | – |
| Leaf color | – | Dark green (15) | – |
| Flower scent | Moderate (16) | High (46) | Very high (6) |
| Fall color | Milky background + Dark brown veins + No spot on the edge (6) | Milky background + Dark brown veins+ Dark brown spot on the edge (6) | Milky/yellow background + Pale brown veins + No spot on the edge (21) |
| Standard color | Milky/yellow background + Gray/brown edge (12) | Milky background + Pale brown edge (18) | Milky/yellow background + Pale brown edge (12) |
| Crest color* | S.A.: yellow/ Pale purple + S.C.: Yellow/Brown (12) | S.A.: Yellow/ purple + S.C.: Yellow/Pale brown (30) | S.A.: Purple/brown + S.C.: Yellow/Pale brown (6) |
| Beard color | – | Yellow (57) | – |
| Patch color | – | Purple (39) | – |
| Stigmatic lip color | Yellow/Pale brown (6) | Yellow/Pale purple (9) | Yellow/purple (9) |
| Anther color | – | Milky/Pale yellow (33) | – |
| Filament color | – | Milky (72) | – |
| Trait | Frequency (No. of accessions) | | |
| | 7 | 8 | 9 | 10 |
| Leaf shape | – | – | – | – |
| Leaf color | – | – | – | – |
| Flower scent | – | – | – | – |
| Fall color | Milky/yellow back-ground + Brown veins + No spot on the edge (30) | Milky/yellow background + Brown veins + Limited spot on the edge (18) | Milky/yellow background + Dark brown veins + Dark brown spot on the edge (9) | – |
| Standard color | Milky background + brown edge (9) | Milky/yellow background + brown edge (9) | Milky background + Dark brown edge (15) | Milky/yellow back-ground + Dark brown edge (3) |
| Crest color* | S.A.: Yellow/ rown + S.C.: Yel-low/brown (9) | S.A.: Milky/brown + S.C.: Dark brown (3) | S.A.: Mainly brown + S.C.: Dark brown (3) | – |

*(Continued)*

**Table 2.** (Continued)

| Trait | Frequency (No. of accessions) | | | |
|---|---|---|---|---|
| | 1 | 2 | | 3 |
| Beard color | Dark yellow (21) | – | – | – |
| Patch color | Dark purple (15) | – | Very dark purple (6) | – |
| Stigmatic lip color | Cream/brown (6) | Mainly brown (3) | – | – |
| Anther color | Milky/yellow (39) | – | – | – |
| Filament color | Milky/violet (9) | – | – | – |

\* S.A.: Style arm; S.C.: Style crest.

The fall color, encompassing combinations of background, vein, and edge spot patterns, is remarkably diverse, with at least ten distinct phenotypic expressions. Notably, the phenotype "Milky/yellow background + Brown veins + No spot on the edge" is the most common (30 accessions), while others like "Milky background + Pale brown veins + No spot" and "Milky/yellow background + Dark brown veins + Dark brown spot" are less frequent. This high phenotypic plasticity may reflect adaptations to visual pollinator cues or environmental gradients, such as UV reflectance or temperature-related pigmentation.

Standard color follows a similarly diverse pattern, with multiple combinations involving milky, yellow, gray, brown, and dark brown tones. The most common types are "Milky background + Gray/brown edge" (21) and "Milky background + Pale brown edge" (18), indicating polymorphism that may influence flower visibility or reproductive success.

Crest color, which encompasses both style arm (S.A.) and style crest (S.C.) pigmentation, exhibits exceptional complexity, with at least nine distinct variants. The most frequently encountered phenotype—S.A.: Yellow/purple + S.C.: Yellow/Pale brown (30 accessions)—highlights the role of floral architecture in reproductive differentiation and pollinator specificity. The presence of rare combinations (e.g., S.A.: Milky/brown + S.C.: Dark brown) also implies potential taxonomic or ecological significance.

Beard color is dominated by the yellow type (57 accessions), while other shades such as pale yellow (24), milky (6), and dark yellow (21) are less common. The yellow pigmentation may serve as a visual nectar guide and may be under strong selective pressure due to its functional role in attracting specific pollinators.

Patch color ranges from very pale purple (3) to very dark purple (6), with pale purple (45) and purple (39) being the most common. This progressive intensity may be genetically controlled and environmentally modulated, possibly contributing to thermoregulation or species recognition by pollinators.

The stigmatic lip color shows pronounced variation, with cream/yellow (57) being predominant, followed by cream/pale brown (12) and several yellow-based phenotypes (e.g., yellow/pale brown, yellow/pale purple, yellow/purple). These variations could influence reproductive isolation or pollen-pistil interactions and suggest ongoing microevolutionary processes within the population.

Anther color is primarily distributed among milky/pale yellow (33) and milky/yellow (39) types, with minor frequencies for milky (30) and white/milky (6). These subtle color variations may be associated with pollen viability, developmental stage, or environmental exposure.

Filament color shows a relatively skewed distribution, with the majority being milky (72), followed by white/milky (24), white (3), and a rare milky/violet variant (9). The dominance of milky coloration points to a possible conserved trait, while the violet hue might indicate introgression or rare mutations.

In conclusion, the qualitative morphological data for *I. meda* accessions exhibit considerable phenotypic richness, reflecting both genetic diversity and environmental responsiveness. The observed variation in floral and vegetative traits

underscores the adaptive complexity of the species and highlights several traits (e.g., flower scent, crest color, fall patterning) as potential markers for ecological differentiation, taxonomic delineation, or breeding programs. The data substantiate the hypothesis that *I. meda* maintains a broad morphological spectrum, likely shaped by local selection pressures and gene flow dynamics.

The diversity identified among the studied *I. meda* accessions is presented in Figs 2–4.

### 3.2. Correlation matrix analysis (CMA)

The simple correlation analysis revealed several statistically significant associations among the examined quantitative morphological traits in *I. meda* accessions, offering insights into their potential co-regulation and functional interdependence (Fig 5).

A strong positive correlation was observed between flower diameter and flower surface ($r = 0.80$, $p < 0.01$), indicating that as the diameter increases, the total visible floral area also expands substantially. This relationship suggests that flower diameter could serve as a reliable proxy for estimating floral display size, which is particularly relevant in ecological studies on pollinator attraction and in ornamental breeding programs.

Similarly, fall length showed significant positive correlations with multiple traits: standard length ($r = 0.72$, $p < 0.01$), patch length ($r = 0.59$, $p < 0.01$), and flower surface ($r = 0.76$, $p < 0.01$). These findings highlight a coordinated development of petal structures, suggesting that accessions with elongated falls also tend to possess longer standards and larger floral surfaces. From a developmental biology perspective, such coordinated growth may reflect shared genetic or hormonal regulation pathways. From a breeding standpoint, these correlations are advantageous, as improving one trait may positively influence others, thereby enhancing overall floral aesthetics and potential reproductive fitness.

The correlation between flower length and flower diameter, though statistically significant ($r = 0.25$, $p < 0.01$), was relatively weak. This indicates a degree of independence in how these two floral dimensions vary among accessions. In practical terms, this suggests that flower length and width may be controlled by partially distinct genetic mechanisms or may be differentially influenced by environmental conditions.

Moreover, spathe length exhibited moderate positive correlations with standard length ($r = 0.50$, $p < 0.01$), bottommost leaf length ($r = 0.50$, $p < 0.01$), and fall width ($r = 0.36$, $p < 0.01$). These associations suggest that spathe length is not only linked to floral organ size but may also reflect aspects of overall vegetative vigor. Such relationships are particularly relevant in taxonomic and ecological studies, as spathe morphology is often used as a diagnostic character in *Iris* species.

Taken together, these findings underscore the complex interplay among floral and vegetative traits in *I. meda*. The significant correlations observed suggest that certain morphological features are co-varying and possibly developmentally integrated. This interdependence should be taken into account in future breeding, conservation, and ecological adaptation studies aimed at improving ornamental value or understanding evolutionary diversification in the genus *Iris*.

Importantly, our results are consistent with those documented by Azimi et al. [32] and Asgari et al. [33], who similarly identified strong associations between floral and vegetative characteristics in wild *Iris* species. This repeated pattern supports the notion that the integration of morphological traits is a stable feature maintained across various environmental conditions and genetic lineages, thereby emphasizing the ecological and evolutionary relevance of these correlations.

In conclusion, the identified morphological correlations here offer a solid basis for more advanced multivariate analyses and highlight the importance of correlation assessments in elucidating phenotypic trait interactions within natural plant populations.

### 3.3. Multiple regression analysis (MRA)

Stepwise multiple linear regression analysis was applied to determine the key predictor variables influencing the dependent variable (Table 3). This analysis was carried out using the linear regression function of the statistical software, where variables were iteratively included or excluded from the model according to a significance criterion of $p \le 0.05$. By applying

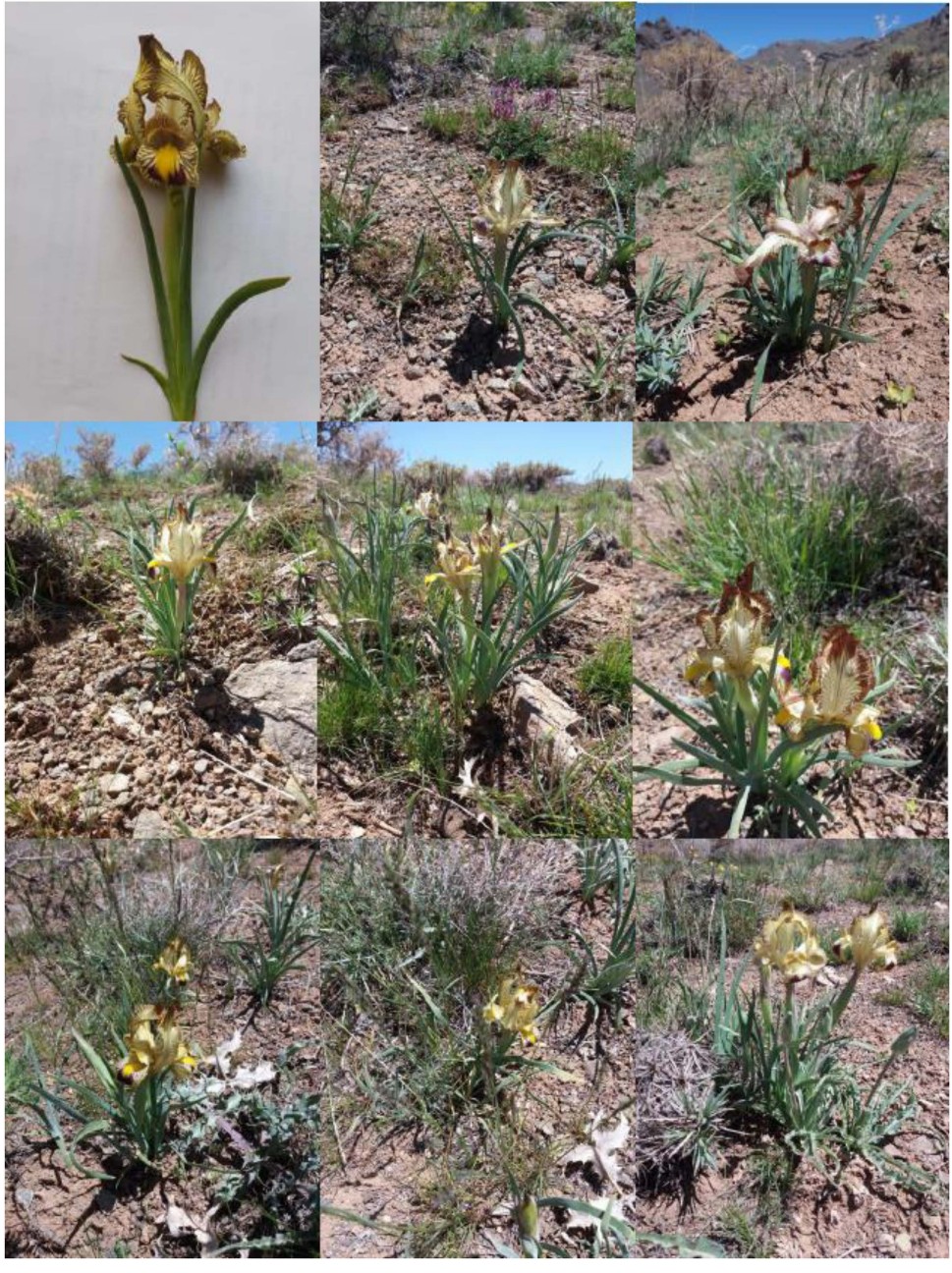

**Fig 2. Variation in bush of *Iris meda* accessions studied.** The images are original and were taken from our plant materials by the first author, Alireza Khaleghi.

this method, a streamlined model was developed, incorporating only those predictors that significantly accounted for variation in the outcome variable [37].

The multiple regression analysis conducted to determine the factors influencing key floral characteristics in *I. meda* revealed several statistically significant relationships that underline the coordinated development of floral and vegetative structures. Among the examined traits, flower diameter was most strongly associated with flower surface ($\beta = 0.87$,

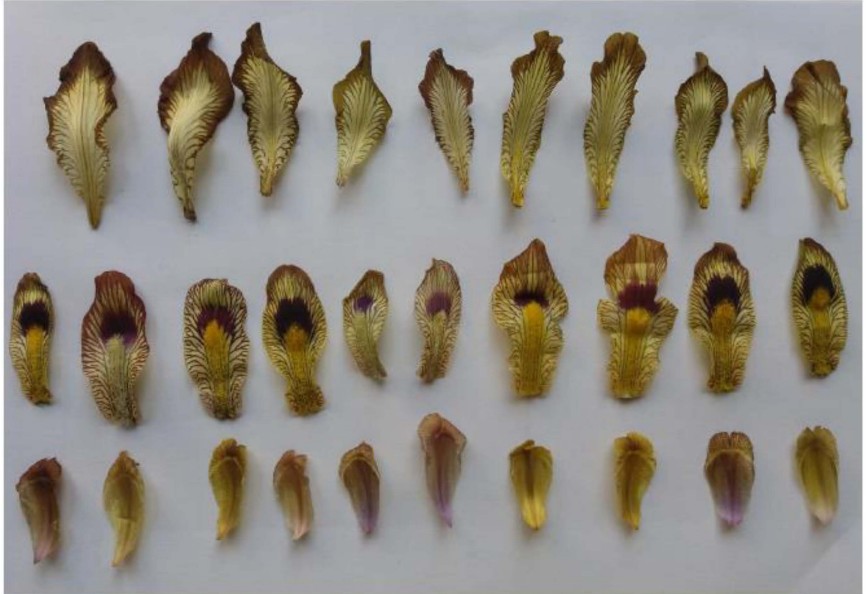

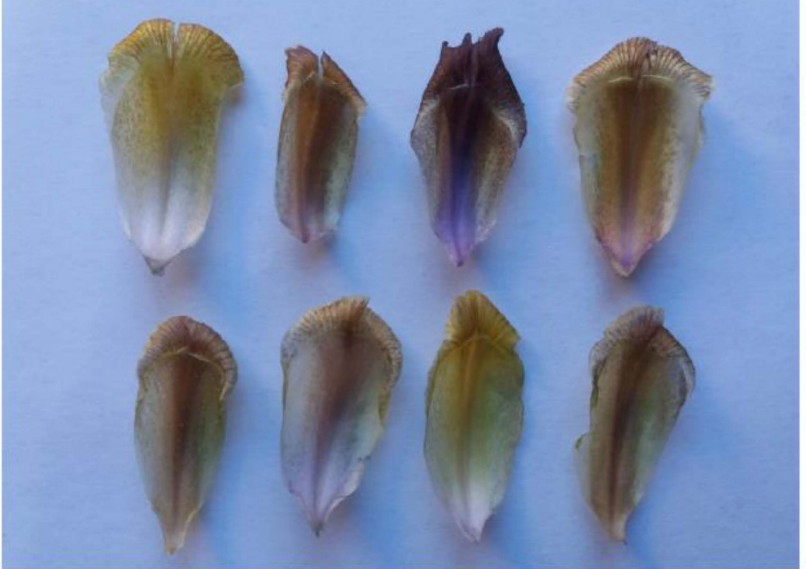

**Fig 3. Variation in fall (sepal), standard (petal), and crest of *Iris meda* accessions studied.** The images are original and were taken from our plant materials by the first author, Alireza Khaleghi.

$p \leq 0.01$), indicating that broader flowers tend to possess greater overall surface area, a relationship likely driven by selection pressures related to pollinator attraction. The positive association with the flower diameter/length ratio ($\beta = 0.51$, $p \leq 0.01$) suggests that more rounded floral forms exhibit greater diameters, reinforcing the functional link between floral shape and display efficiency. Beard length ($\beta = 0.02$, $p \leq 0.01$) and bottommost leaf width ($\beta = 0.02$, $p \leq 0.01$) also contributed positively, although their relatively low coefficients indicate minor influences. In contrast, flower length exhibited a negative association with flower diameter ($\beta = -0.15$, $p \leq 0.01$), suggesting a structural trade-off whereby longer flowers are not necessarily broader. Similarly, leaf number showed a weak but significant negative effect ($\beta = -0.01$, $p \leq 0.01$), implying

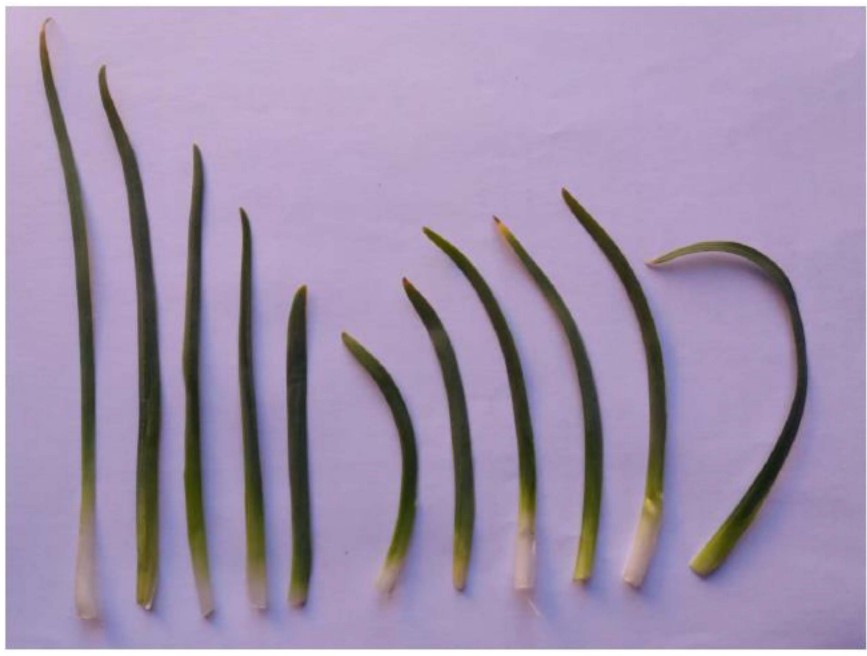

**Fig 4. Variation in bottommost leaf of *Iris meda* accessions studied.** The images are original and were taken from our plant materials by the first author, Alireza Khaleghi.

that higher vegetative complexity may slightly constrain floral expansion. A weak positive relationship with style flap length ($\beta = 0.01$, $p \leq 0.05$) was also observed.

Regarding fall length, beard length emerged as the most influential predictor ($\beta = 0.63$, $p \leq 0.01$), highlighting the developmental interdependence of petal appendages involved in visual signaling and potentially guiding pollinators. Fall width also showed a strong positive effect ($\beta = 0.43$, $p \leq 0.01$), further emphasizing the coordinated development of floral symmetry elements. Additional positive associations were found with flower diameter ($\beta = 0.19$, $p \leq 0.01$), spathe width ($\beta = 0.10$, $p \leq 0.01$), style arm length ($\beta = 0.13$, $p \leq 0.01$), bottommost leaf width ($\beta = 0.16$, $p \leq 0.01$), and patch width ($\beta = 0.12$, $p \leq 0.01$), suggesting that both vegetative and floral structures contribute to fall development. Conversely, filament length ($\beta = -0.19$, $p \leq 0.01$) and peduncle diameter ($\beta = -0.33$, $p \leq 0.01$) were negatively associated with fall length, which may reflect structural compensation mechanisms within the inflorescence. Notably, patch length was also negatively correlated ($\beta = -0.10$, $p \leq 0.05$), implying variation in the proportional contributions of pigmentation and structural components to fall elongation. A moderate negative relationship with leaf number ($\beta = -0.13$, $p \leq 0.01$) again indicates a possible trade-off between vegetative density and floral elongation.

Spathe length was significantly influenced by a combination of reproductive and vegetative traits. Beard length was the strongest positive predictor ($\beta = 0.53$, $p \leq 0.01$), followed by bottommost leaf length ($\beta = 0.35$, $p \leq 0.01$), implying a developmental parallelism between bract elongation and vegetative robustness. The association with peduncle diameter ($\beta = 0.25$, $p \leq 0.01$) suggests that individuals with thicker peduncles may support larger spathes, potentially enhancing floral protection and display. The patch surface to fall width ratio exhibited a significant negative effect on spathe length ($\beta = -0.30$, $p \leq 0.01$). This suggests that an increased proportion of pigmented area relative to fall width may constrain the development of the spathe. Such a pattern could reflect a trade-off in resource allocation between visually conspicuous floral elements and structural protective organs. The reduction in spathe length in flowers with higher visual contrast might indicate a developmental prioritization toward pollinator attraction at the expense

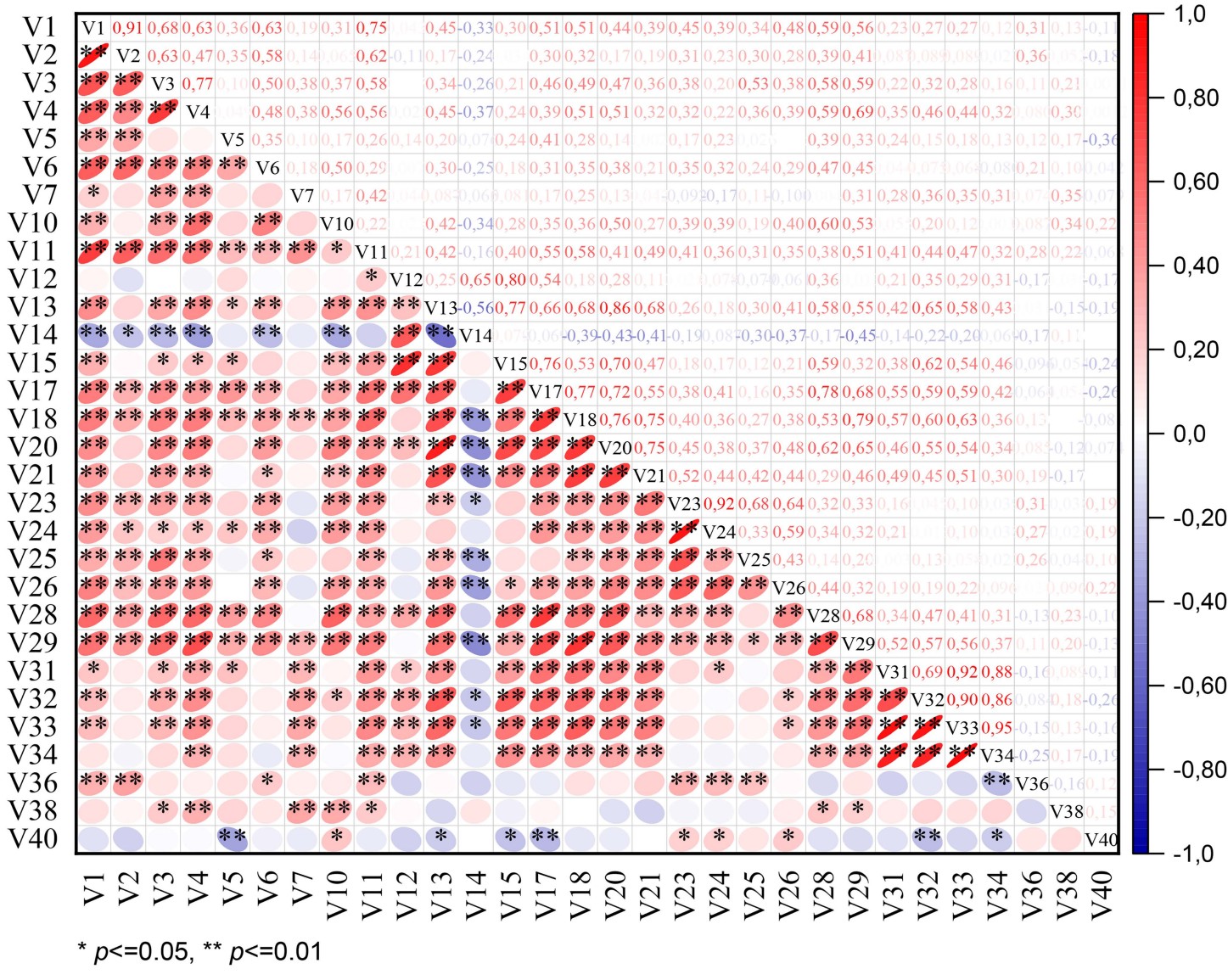

**Fig 5. Simple correlation matrix illustrating the relationships among quantitative morphological traits evaluated in the *Iris meda* accessions.**
Refer to Table 1 for abbreviations.

of protective structures. This inverse relationship may also highlight a broader ecological or evolutionary strategy, wherein enhanced pigmentation and visual signaling potentially downregulate the investment in bract elongation, possibly as an adaptive response to environmental or reproductive pressures. A noteworthy negative correlation was identified with stem length ($\beta = -0.72$, $p \le 0.01$), indicating a potential trade-off between internodal elongation and protective bract development. Other traits such as anther length ($\beta = 0.20$, $p \le 0.01$), spathe width ($\beta = 0.19$, $p \le 0.01$), and style flap length ($\beta = 0.19$, $p \le 0.01$) contributed positively, while the flower diameter/length ratio ($\beta = -0.21$, $p \le 0.01$) had a significant negative effect. These associations reinforce the complex interplay between floral form, function, and structural allocation, suggesting that both reproductive structures and geometric proportions influence spathe length in a coordinated yet contrasting manner.

**Table 3. Morphological traits associated with floral characteristics in *Iris meda* accessions as identified through multiple regression analysis and corresponding coefficients.**

| Dependent character | Independent character | r | r² | β | t-value | p-value |
|---|---|---|---|---|---|---|
| Flower diameter | Flower surface | 0.804[a] | 0.65 | 0.87 | 25.23 | 0.01*[a] |
| | Flower diameter/length | 0.999[b] | 1.00 | 0.51 | 19.50 | 0.01* [a] |
| | Bottommost leaf width | 0.999[c] | 1.00 | 0.02 | 4.69 | 0.01* |
| | Beard length | 0.999[d] | 1.00 | 0.02 | 4.08 | 0.01* [a] |
| | Flower length | 0.999[e] | 1.00 | −0.15 | −3.81 | 0.01* [a] |
| | Leaf number | 0.999[f] | 1.00 | −0.01 | −2.64 | 0.01* [a] |
| | Style flap length | 0.999[g] | 1.00 | 0.01 | 2.20 | 0.05** |
| Fall length | Beard length | 0.776[a] | 0.60 | 0.63 | 13.06 | 0.01* [a] |
| | Fall width | 0.881[b] | 0.78 | 0.43 | 10.49 | 0.01* [a] |
| | Flower diameter | 0.928[c] | 0.86 | 0.19 | 5.71 | 0.01* [a] |
| | Filament length | 0.937[d] | 0.88 | −0.19 | −6.34 | 0.01* [a] |
| | Peduncle diameter | 0.945[e] | 0.89 | −0.33 | −7.11 | 0.01* [a] |
| | Spathe width | 0.951[f] | 0.91 | 0.10 | 2.62 | 0.01* [a] |
| | Style arm length | 0.954[g] | 0.91 | 0.13 | 3.72 | 0.01* [a] |
| | Bottommost leaf width | 0.960[h] | 0.92 | 0.16 | 4.50 | 0.01* [a] |
| | Leaf number | 0.963[i] | 0.93 | −0.13 | −3.61 | 0.01* [a] |
| | Patch width | 0.965[j] | 0.93 | 0.12 | 3.16 | 0.01* [a] |
| | Patch length | 0.967[k] | 0.94 | −0.10 | −2.01 | 0.05** [a] |
| Spathe length | Beard length | 0.602[a] | 0.36 | 0.53 | 8.27 | 0.01* [a] |
| | Bottommost leaf length | 0.665[b] | 0.44 | 0.35 | 5.45 | 0.01* [a] |
| | Stem length | 0.708[c] | 0.50 | −0.72 | −9.70 | 0.01* |
| | Peduncle diameter | 0.770[d] | 0.59 | 0.25 | 3.43 | 0.01* [a] |
| | Patch surface/fall width | 0.810[e] | 0.66 | −0.30 | −5.23 | 0.01*[a] |
| | Style flap length | 0.836[f] | 0.70 | 0.19 | 3.44 | 0.01* |
| | Anther length | 0.853[g] | 0.73 | 0.20 | 3.96 | 0.01* [a] |
| | Flower diameter/length | 0.866[h] | 0.75 | −0.21 | −4.03 | 0.01* [a] |
| | Spathe width | 0.862[i] | 0.74 | 0.19 | 2.58 | 0.01* |

r Correlation coefficient, r² Coefficient of determination, β Standardized beta coefficients.

*, **. Regression correlation is significant at $p \leq 0.05$ and $p \leq 0.01$ levels, respectively.

[a]Multiple regression analysis correlations supported by the correlation matrix analysis.

Overall, the multivariate relationships observed in this analysis point to a tightly integrated pattern of morphological variation in *I. meda*, where floral dimensions are intricately linked with both reproductive traits and vegetative structures. These coordinated changes suggest that selection may favor specific trait combinations that optimize reproductive success, mechanical support, and ecological adaptation. The identification of strong and significant predictors for key floral characteristics offers valuable markers for future taxonomic studies, ornamental breeding programs, and ecological assessments of phenotypic plasticity in natural populations.

In the assessment of morphological characteristics of *I. meda*, flower diameter, fall length, and spathe length emerged as the three most influential traits. Flower diameter serves as a primary determinant of ornamental value, as it significantly affects the plant's visual appeal. Larger flowers tend to be more aesthetically pleasing and exhibit higher genotypic diversity, rendering this trait particularly advantageous for selection and breeding initiatives. Additionally, more prominent floral structures are often associated with increased attractiveness to pollinators, potentially enhancing reproductive efficiency [38]. Fall length, which denotes the length of the outer tepals, is a vital morphological marker in the taxonomy of

*Iris* species and is integral to the overall floral design. It also influences the symmetrical appearance of the bloom and may act as a physical platform for pollinators, thereby impacting pollination success [39]. Spathe length, which encloses and protects the floral bud before anthesis, plays a crucial role in environmental adaptation. Owing to its measurability in early developmental stages, this trait offers a practical criterion for early-stage selection in breeding programs [40].

### 3.4. Principal component analysis (PCA)

According to Kaiser's criterion, this study retained only those principal components (PCs) with eigenvalues exceeding 1.0 for subsequent interpretation, as components with eigenvalues below this threshold account for less variance than an individual original variable and are thus deemed statistically insignificant [41].

Accordingly, the first 11 principal components with eigenvalues exceeding 1 collectively accounted for 83.72% of the total variance, indicating a substantial dimensional reduction while retaining most of the information embedded in the original dataset (Table 4).

The first PC1 alone explained 28.86% of the total variance and was predominantly defined by flower length (0.89), standard length (0.86), flower surface (0.70), standard width (0.63), and fall width (0.62). These variables, mainly floral morphological features, suggest that PC1 captures variation associated with floral size and architecture. The strong loadings indicate a coordinated variation among these traits, which may be reflective of genetic or developmental linkages.

The second PC2, accounting for 11.15% of the variance, was mainly influenced by stem length (0.94), plant length (0.86), stem diameter (0.76), and spathe width (0.63). This component appears to reflect vegetative vigor and plant stature, separating accessions based on overall plant size and robustness. The high loadings on both stem and spathe attributes highlight the potential correlation between shoot elongation and reproductive structures.

The third PC3 explained 8.86% of the total variance and was strongly associated with patch surface/fall width ratio (0.95), patch surface (0.92), patch width (0.92), and patch length (0.75). These traits relate to ornamental patterning and pigmentation zones, indicating that PC3 captures the variation in floral pigmentation and visual display, which may have ecological and taxonomic implications.

Collectively, the first three components explained 48.87% of the total variance, effectively summarizing nearly half of the phenotypic variability in the dataset. This demonstrates the presence of distinct, biologically meaningful trait groupings within the accessions. PC1 highlights floral dimensional traits, PC2 reflects vegetative growth parameters, and PC3 encapsulates pigmentation and visual patterning. These findings suggest a multi-dimensional trait structure among the studied accessions, underscoring the complex interplay between floral morphology, plant architecture, and ornamental characteristics. Such insights are valuable for both taxonomic discrimination and breeding efforts focused on morphological and aesthetic criteria.

Similar observations have been documented in earlier studies involving *Iris* species. For instance, research conducted by Sapir et al. [20] in Israel, Azimi et al. [31] in Iran, Bo et al. [42] in China, Boltenkov et al. [43] in Russia, and Ghorbani et al. [34] in Iran revealed that the first three principal components accounted for over 40% of the total variance in their respective analyses. These consistent patterns lend further credibility to the present results and highlight the broader applicability of multivariate morphological variation trends observed within the *Iris* genus.

In the principal component biplot (PC1 vs. PC2), which summarizes the multivariate relationships among *I. meda* accessions based on morphological traits, most accessions clustered within the 95% confidence ellipse, indicating a general morphological similarity among the majority of the studied individuals. However, certain accessions—specifically '*I. meda*-10', '*I. meda*-23', '*I. meda*-27', '*I. meda*-59', '*I. meda*-81', '*I. meda*-82', '*I. meda*-85', '*I. meda*-93', '*I. meda*-95', '*I. meda*-99', '*I. meda*-107'—were positioned outside the ellipse, suggesting that they exhibit notable morphological divergence from the core group (Fig 6). The displacement of these accessions along the principal components may be attributed to distinct variations in one or more key morphological traits such as flower dimensions, leaf architecture, or plant stature, which have higher loading scores on PC1 or PC2. Their outlier status potentially reflects underlying genetic

**Table 4. Principal component eigenvalues derived from PCA of morphological variables in *Iris meda* accessions.**

| Trait | Component | | | | | | | | | | |
|---|---|---|---|---|---|---|---|---|---|---|---|
| | 1 | 2 | 3 | 4 | 5 | 6 | 7 | 8 | 9 | 10 | 11 |
| Plant length | 0.24 | **0.86**[a] | 0.10 | 0.26 | 0.01 | 0.00 | 0.00 | 0.07 | 0.12 | −0.01 | −0.06 |
| Stem length | −0.03 | **0.94**[a] | −0.01 | 0.12 | 0.11 | −0.07 | 0.07 | −0.04 | 0.07 | −0.03 | 0.01 |
| Stem diameter | 0.27 | **0.76**[a] | 0.12 | 0.07 | −0.14 | −0.06 | −0.22 | 0.24 | −0.29 | 0.04 | 0.01 |
| Peduncle diameter | 0.36 | 0.56 | 0.27 | 0.11 | −0.18 | −0.19 | −0.17 | 0.37 | −0.21 | −0.12 | −0.11 |
| Leaf number | 0.06 | 0.33 | 0.11 | 0.15 | −0.14 | 0.10 | 0.60 | −0.02 | 0.39 | −0.09 | −0.15 |
| Bottommost leaf length | 0.31 | 0.58 | −0.15 | 0.18 | −0.06 | −0.12 | 0.34 | 0.09 | −0.12 | −0.13 | −0.17 |
| Bottommost leaf width | 0.09 | 0.23 | 0.24 | −0.25 | 0.17 | −0.01 | 0.13 | **0.71**[a] | −0.10 | 0.17 | −0.08 |
| Leaf shape | 0.13 | −0.13 | 0.29 | 0.13 | −0.08 | 0.05 | −0.14 | −0.02 | −0.14 | 0.01 | **0.63**[a] |
| Leaf color | 0.08 | 0.07 | −0.05 | 0.28 | 0.05 | −0.28 | 0.14 | 0.23 | 0.04 | **0.76**[a] | 0.03 |
| Spathe length | 0.53 | 0.08 | −0.08 | 0.33 | −0.38 | −0.16 | 0.12 | 0.39 | −0.20 | −0.18 | −0.27 |
| Spathe width | 0.21 | **0.63**[a] | 0.32 | 0.25 | 0.14 | 0.19 | −0.05 | 0.32 | 0.12 | 0.24 | −0.07 |
| Flower diameter | 0.25 | −0.03 | 0.18 | −0.01 | −0.10 | **0.93**[a] | 0.11 | 0.01 | −0.06 | −0.04 | 0.01 |
| Flower length | **0.89**[a] | 0.17 | 0.31 | 0.06 | 0.04 | 0.01 | −0.05 | −0.10 | 0.10 | −0.03 | −0.03 |
| Flower diameter/length | −0.48 | −0.17 | −0.09 | −0.06 | −0.10 | **0.78**[a] | 0.09 | 0.06 | −0.14 | −0.02 | 0.05 |
| Flower surface | **0.70**[a] | 0.08 | 0.29 | 0.04 | −0.06 | 0.61 | 0.05 | −0.05 | 0.05 | −0.04 | −0.01 |
| Flower scent | 0.42 | 0.44 | 0.16 | −0.22 | −0.03 | 0.06 | 0.28 | −0.19 | 0.21 | 0.41 | 0.12 |
| Fall length | 0.56 | 0.34 | 0.36 | 0.22 | −0.24 | 0.34 | 0.27 | 0.03 | 0.02 | 0.13 | 0.24 |
| Fall width | **0.62**[a] | 0.34 | 0.37 | 0.22 | 0.01 | −0.05 | 0.21 | 0.09 | −0.05 | 0.29 | 0.18 |
| Fall color | −0.23 | 0.10 | 0.17 | −0.21 | **0.80**[a] | −0.04 | 0.01 | −0.01 | 0.12 | 0.20 | −0.01 |
| Standard length | **0.86**[a] | 0.19 | 0.24 | 0.24 | −0.08 | 0.01 | −0.02 | −0.01 | −0.03 | 0.06 | 0.06 |
| Standard width | **0.63**[a] | 0.16 | 0.29 | 0.34 | 0.18 | −0.05 | −0.12 | −0.09 | 0.01 | 0.43 | 0.06 |
| Standard color | 0.00 | −0.03 | −0.03 | −0.26 | 0.39 | −0.01 | −0.06 | −0.12 | 0.13 | 0.03 | **0.73**[a] |
| Crest length | 0.21 | 0.28 | −0.01 | **0.88**[a] | −0.01 | 0.01 | −0.09 | −0.04 | −0.07 | 0.09 | 0.01 |
| Style arm length | 0.12 | 0.17 | 0.02 | **0.92**[a] | −0.13 | 0.05 | 0.09 | −0.06 | 0.05 | 0.11 | 0.01 |
| Style crest length | 0.28 | 0.37 | −0.05 | 0.38 | 0.20 | −0.06 | −0.38 | 0.01 | −0.24 | 0.00 | −0.01 |
| Crest width | 0.32 | 0.21 | 0.13 | **0.63**[a] | −0.09 | −0.14 | −0.33 | 0.04 | 0.16 | −0.09 | −0.04 |
| Crest color | 0.49 | 0.21 | 0.29 | 0.17 | −0.22 | −0.04 | −0.06 | 0.05 | 0.34 | −0.06 | 0.39 |
| Beard length | 0.49 | 0.44 | 0.23 | 0.19 | −0.54 | 0.15 | 0.13 | 0.04 | −0.05 | −0.22 | 0.01 |
| Beard width | 0.48 | 0.43 | 0.35 | 0.18 | −0.23 | −0.27 | 0.30 | 0.26 | −0.09 | 0.00 | 0.23 |
| Beard color | 0.04 | −0.09 | 0.12 | −0.20 | 0.03 | 0.10 | **0.85**[a] | 0.03 | −0.04 | 0.07 | −0.08 |
| Patch width | 0.16 | 0.10 | **0.92**[a] | 0.11 | −0.06 | −0.01 | 0.12 | 0.01 | 0.03 | 0.15 | 0.07 |
| Patch length | 0.48 | 0.11 | **0.75**[a] | −0.08 | 0.09 | 0.12 | 0.05 | 0.20 | 0.00 | −0.14 | 0.12 |
| Patch surface | 0.33 | 0.10 | **0.92**[a] | 0.02 | 0.04 | 0.05 | 0.06 | 0.11 | 0.01 | 0.03 | 0.08 |
| Patch surface/fall width | 0.16 | 0.00 | **0.95**[a] | −0.08 | 0.02 | 0.11 | −0.01 | 0.09 | 0.04 | −0.10 | 0.03 |
| Patch color | 0.04 | 0.25 | 0.03 | 0.08 | −0.20 | −0.27 | 0.15 | 0.30 | 0.10 | **−0.62**[a] | 0.04 |
| Style flap length | 0.06 | 0.30 | −0.24 | 0.30 | 0.59 | −0.08 | 0.17 | 0.01 | −0.12 | −0.01 | −0.12 |
| Style flap color | 0.00 | −0.22 | −0.02 | 0.12 | 0.28 | −0.27 | 0.01 | 0.13 | **0.82**[a] | 0.07 | −0.08 |
| Anther length | −0.17 | 0.08 | 0.12 | 0.05 | −0.26 | 0.07 | −0.07 | **0.77**[a] | 0.15 | −0.11 | −0.03 |
| Anther color | 0.33 | 0.36 | 0.30 | −0.21 | −0.26 | −0.06 | −0.27 | 0.04 | 0.32 | −0.04 | −0.37 |
| Filament length | −0.15 | −0.27 | −0.07 | 0.40 | −0.01 | −0.15 | −0.22 | 0.17 | −0.52 | 0.13 | −0.27 |
| Filament color | 0.13 | −0.16 | 0.08 | −0.05 | **0.83**[a] | −0.11 | −0.14 | −0.14 | 0.13 | −0.03 | 0.29 |
| *Eigenvalue* | *11.83* | *4.57* | *3.63* | *2.89* | *2.43* | *2.03* | *1.86* | *1.55* | *1.35* | *1.17* | *1.01* |
| *Component degree of significance* | *** | *** | *** | *** | *** | *** | ** | *** | ** | *** | *** |
| *Variance (%)* | *28.86* | *11.15* | *8.86* | *7.05* | *5.92* | *4.94* | *4.53* | *3.78* | *3.30* | *2.86* | *2.46* |
| *∑ variance (%)* | *28.86* | *40.01* | *48.87* | *55.92* | *61.84* | *66.78* | *71.31* | *75.10* | *78.40* | *81.26* | *83.72* |

[a]Bold values indicate the characteristics that most influence each PC (Eigenvalues are significant ≥ 0.62). Component degree of significance: *$p < 0.05$, **$p < 0.01$.

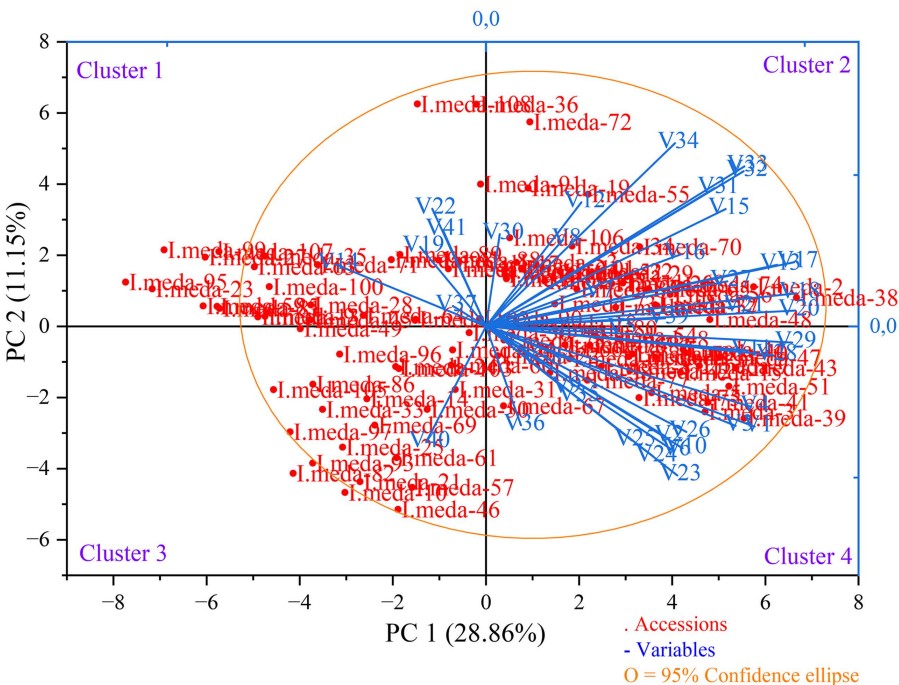

**Fig 6. Principal component biplot (PC1 vs. PC2) displaying the distribution of *Iris meda* accessions concerning morphological variables.** Abbreviations are provided in Table 1.

variability or adaptation to specific microecological conditions within the species' native range [44,45]. The identification of such morphologically divergent genotypes is particularly valuable for breeding, conservation, and taxonomic studies, as they may harbor unique traits of ecological or ornamental significance.

The PC scores of *I. meda* accessions ranged from −9.87 ('*I. meda*-66') to 36.35 ('*I. meda*-59'), indicating considerable variation in multivariate morphological composition among genotypes (Table 5). The accessions with the highest PC scores were '*I. meda*-59' (36.35), '*I. meda*-27' (34.63), '*I. meda*-82' (34.44), '*I. meda*-81' (33.70), '*I. meda*-10' (33.45), '*I. meda*-85' (33.30), '*I. meda*-95' (33.27), '*I. meda*-23' (32.93), '*I. meda*-93' (32.54), '*I. meda*-36' (32.22), '*I. meda*-7' (31.64), '*I. meda*-3' (31.44), '*I. meda*-107' (30.53), and '*I. meda*-99' (30.16). Among these, '*I. meda*-10', '*I. meda*-23', '*I. meda*-27', '*I. meda*-59', '*I. meda*-81', '*I. meda*-82', '*I. meda*-85', '*I. meda*-93', '*I. meda*-95', '*I. meda*-99', and '*I. meda*-107' were located outside the 95% confidence ellipse in the PCA biplot, highlighting their pronounced morphological divergence from the central cluster. Furthermore, '*I. meda*-36' exhibited the largest flower diameter (71.23 mm), '*I. meda*-7' had the longest fall length (54.63 mm), and '*I. meda*-3' possessed the greatest spathe length (71.88 mm). These findings from the PCA scores analysis reinforce the trends observed in the descriptive statistics and biplot visualization, confirming the presence of substantial morphological variability among accessions. Unlike previous studies where PC score analyses were not reported, the integration of this approach in the current study enables a more comprehensive and robust interpretation of inter-accession morphological differentiation. The combined use of PCA and CMA has been effectively employed to assess genetic diversity and classify accessions in date palm (*Phoenix dactylifera* L.) [46].

### 3.5. Hierarchical cluster analysis (HCA)

Fig 7 illustrates the dendrogram generated through hierarchical cluster analysis (HCA) based on the quantitative morphological traits of *I. meda* accessions. S1 Table provides detailed information on the stepwise clustering

**Table 5. Principal component scores of different *Iris meda* accessions.**

| Accession | PC1 | PC2 | PC3 | PC4 | PC5 | PC6 | PC7 | PC8 | PC9 | Composite score |
|---|---|---|---|---|---|---|---|---|---|---|
| 'I. meda-1' | −0.50 | 1.80 | 0.93 | 0.39 | −1.38 | −1.38 | −1.77 | 1.46 | 0.40 | −0.05 |
| 'I. meda-2' | 0.83 | −1.92 | 1.88 | 1.33 | −1.15 | −1.27 | −1.27 | −0.78 | 0.10 | −2.25 |
| 'I. meda-3' | 3.58 | 3.23 | 4.03 | 2.85 | 3.23 | 3.42 | 3.64 | 4.46 | 3.00 | 31.44 |
| 'I. meda-4' | 0.06 | 0.37 | −1.81 | 0.43 | −1.32 | −1.74 | 1.80 | 1.86 | 1.23 | 0.88 |
| 'I. meda-5' | −0.78 | −1.61 | 0.74 | −0.24 | −1.51 | −0.02 | −1.86 | 1.64 | −0.96 | −4.60 |
| 'I. meda-6' | 0.65 | −0.75 | 0.08 | 0.19 | −1.26 | 1.88 | 1.10 | 1.76 | 1.58 | 5.23 |
| 'I. meda-7' | 3.99 | 4.80 | 2.72 | 2.99 | 2.61 | 3.31 | 3.47 | 3.18 | 4.57 | 31.64 |
| 'I. meda-8' | −0.57 | −0.88 | 0.17 | −1.44 | 1.21 | −1.70 | 1.95 | 1.09 | −1.21 | −1.38 |
| 'I. meda-9' | −1.98 | 1.26 | 0.83 | 0.92 | 1.09 | −1.70 | −0.57 | −1.54 | 1.45 | −0.24 |
| 'I. meda-10' | 4.06 | 3.33 | 2.66 | 3.28 | 3.31 | 4.32 | 4.09 | 4.72 | 3.68 | 33.45 |
| 'I. meda-11' | −1.52 | 0.85 | 1.04 | 0.25 | 1.08 | −0.02 | 0.09 | −0.29 | −1.90 | −0.42 |
| 'I. meda-12' | −1.57 | −1.87 | 0.55 | −0.74 | 0.03 | 1.63 | −1.00 | −0.36 | 1.02 | −2.31 |
| 'I. meda-13' | −1.08 | −1.69 | −0.84 | −1.36 | 1.72 | 1.23 | 0.53 | 1.49 | 1.21 | 1.21 |
| 'I. meda-14' | −1.25 | 1.57 | 0.16 | 1.23 | 1.58 | −0.73 | −1.56 | −1.09 | −0.29 | −0.38 |
| 'I. meda-15' | 1.27 | 1.44 | −1.97 | 0.04 | −0.33 | −1.11 | −1.52 | −0.65 | 1.77 | −1.06 |
| 'I. meda-16' | −0.71 | 0.08 | 0.81 | −0.55 | 1.89 | 1.85 | −0.99 | −0.01 | −0.80 | 1.57 |
| 'I. meda-17' | −0.86 | −1.85 | 0.44 | 0.01 | −1.79 | −0.89 | 1.63 | −1.04 | −1.42 | −5.77 |
| 'I. meda-18' | −0.04 | 1.94 | −1.03 | 0.69 | 1.05 | −1.05 | 0.91 | −0.53 | 0.53 | 2.47 |
| 'I. meda-19' | 0.53 | 0.14 | −1.64 | 1.34 | −0.72 | −1.25 | −1.84 | 0.36 | 0.71 | −2.37 |
| 'I. meda-20' | −1.93 | 0.05 | −1.09 | 0.58 | −1.30 | 0.76 | −0.45 | 1.75 | −1.45 | −3.08 |
| 'I. meda-21' | −0.64 | −1.55 | 1.70 | 1.51 | −0.97 | 0.64 | 1.27 | 0.22 | 0.12 | 2.30 |
| 'I. meda-22' | −1.03 | −1.63 | 1.59 | 1.60 | 0.53 | −0.64 | −0.60 | 0.90 | 1.59 | 2.31 |
| 'I. meda-23' | 4.72 | 4.45 | 4.11 | 2.71 | 2.90 | 4.75 | 4.02 | 2.52 | 2.75 | 32.93 |
| 'I. meda-24' | 0.65 | −1.98 | −1.36 | 0.19 | 0.77 | 0.61 | −1.10 | 0.85 | −1.05 | −2.42 |
| 'I. meda-25' | −0.70 | 0.99 | 0.60 | 1.40 | 0.63 | 0.27 | −1.63 | −0.53 | −0.94 | 0.09 |
| 'I. meda-26' | −1.02 | 1.89 | −0.43 | 1.57 | 0.52 | 1.18 | 0.01 | 0.31 | −0.03 | 4.00 |
| 'I. meda-27' | 2.99 | 4.31 | 3.20 | 2.56 | 4.11 | 2.94 | 4.85 | 4.88 | 4.79 | 34.63 |
| 'I. meda-28' | −0.52 | −1.94 | 1.71 | −0.29 | 1.87 | 1.85 | 1.41 | −0.82 | −0.46 | 2.81 |
| 'I. meda-29' | 1.40 | −0.73 | −1.32 | 0.23 | 1.74 | 0.78 | 0.28 | −1.61 | 0.46 | 1.23 |
| 'I. meda-30' | 1.96 | −1.44 | 0.07 | 1.51 | 0.96 | 0.79 | 0.81 | −0.56 | −0.83 | 3.27 |
| 'I. meda-31' | 1.24 | 1.24 | 1.47 | 1.65 | 0.05 | 0.01 | 1.19 | 0.60 | 0.81 | 8.26 |
| 'I. meda-32' | 1.18 | 1.56 | −0.65 | −0.50 | −1.62 | 0.31 | −1.86 | −0.14 | 0.17 | −1.55 |
| 'I. meda-33' | −0.85 | 0.36 | −1.88 | −1.85 | 1.29 | −0.56 | −1.49 | 0.09 | 1.08 | −3.81 |
| 'I. meda-34' | −1.14 | 0.49 | −1.66 | −1.79 | 0.13 | 0.16 | 0.55 | 0.90 | 1.90 | −0.46 |
| 'I. meda-35' | 0.07 | −0.71 | 1.18 | −0.92 | −0.24 | −1.69 | −1.90 | 1.85 | 1.34 | −1.02 |
| 'I. meda-36' | 4.24 | 3.52 | 2.93 | 2.89 | 3.13 | 3.87 | 4.29 | 4.15 | 3.20 | 32.22 |
| 'I. meda-37' | 1.82 | 0.95 | 0.22 | 0.45 | −0.32 | −1.01 | −0.58 | 1.03 | −1.94 | 0.62 |
| 'I. meda-38' | −1.54 | −1.82 | −1.84 | 1.42 | 0.81 | −0.10 | −1.61 | −0.03 | −0.11 | −4.82 |
| 'I. meda-39' | −1.31 | −0.26 | −0.41 | 0.46 | 0.54 | −1.82 | −0.50 | 0.50 | 0.01 | −2.79 |
| 'I. meda-40' | 1.43 | 0.63 | −1.35 | −1.72 | 0.57 | −1.89 | 0.34 | 1.76 | 0.30 | 0.07 |
| 'I. meda-41' | −0.45 | 0.57 | −0.17 | 0.18 | 1.77 | −0.46 | 1.84 | 1.62 | −1.22 | 3.68 |
| 'I. meda-42' | −1.72 | −1.60 | −1.93 | −1.62 | 0.73 | −1.72 | −0.72 | 1.38 | −1.91 | −9.11 |
| 'I. meda-43' | 1.26 | −0.87 | −1.53 | 0.79 | 0.52 | 1.51 | 0.94 | 1.21 | −0.87 | 2.96 |
| 'I. meda-44' | −1.29 | 1.00 | 1.23 | 1.96 | −0.35 | −0.51 | 1.11 | −0.64 | 1.72 | 4.23 |
| 'I. meda-45' | 1.43 | −0.28 | 1.00 | 1.02 | −1.59 | 1.61 | 0.02 | 1.31 | −0.72 | 3.80 |
| 'I. meda-46' | 1.58 | −0.44 | −1.96 | 1.62 | −1.63 | −0.72 | 1.80 | 1.80 | 0.29 | 2.34 |

*(Continued)*

**Table 5.** (Continued)

| Accession | PC1 | PC2 | PC3 | PC4 | PC5 | PC6 | PC7 | PC8 | PC9 | Composite score |
|---|---|---|---|---|---|---|---|---|---|---|
| 'I. meda-47' | 0.53 | −0.21 | −0.83 | −0.69 | 0.69 | 1.01 | 1.17 | 1.16 | −1.64 | 1.19 |
| 'I. meda-48' | −0.02 | −1.77 | 0.20 | −0.23 | 1.55 | −0.60 | −1.53 | −1.43 | 1.05 | −2.78 |
| 'I. meda-49' | 0.47 | −1.60 | −1.66 | 0.80 | −1.71 | 1.29 | 0.82 | −1.67 | −1.66 | −4.92 |
| 'I. meda-50' | 1.95 | −0.50 | −0.52 | 1.25 | 1.79 | 1.94 | 1.01 | −0.49 | −1.67 | 4.76 |
| 'I. meda-51' | 1.11 | 0.23 | −0.30 | 1.63 | −1.56 | −0.03 | −1.95 | −0.13 | −1.77 | −2.77 |
| 'I. meda-52' | −1.52 | −1.53 | 0.60 | 0.98 | 0.33 | 1.85 | −0.50 | −0.86 | 1.47 | 0.82 |
| 'I. meda-53' | −1.11 | 1.85 | −1.95 | 1.88 | −1.83 | 1.56 | 0.11 | 1.97 | −1.70 | 0.78 |
| 'I. meda-54' | 0.22 | 1.88 | 0.09 | 0.52 | 0.78 | −0.18 | 0.51 | 0.34 | 1.60 | 5.76 |
| 'I. meda-55' | −1.82 | −0.88 | 1.80 | 1.56 | −0.18 | 0.48 | −0.89 | −1.25 | −0.15 | −1.33 |
| 'I. meda-56' | −0.59 | 0.33 | −1.69 | 1.90 | 1.94 | 0.79 | 0.14 | −0.76 | 1.26 | 3.32 |
| 'I. meda-57' | 0.74 | −1.35 | 1.64 | 1.29 | 1.80 | 0.90 | 0.45 | −0.33 | 1.73 | 6.87 |
| 'I. meda-58' | 1.46 | −1.82 | −1.89 | −0.49 | 1.24 | 1.95 | −1.40 | 0.38 | −0.48 | −1.05 |
| 'I. meda-59' | 4.92 | 4.61 | 4.60 | 3.67 | 3.54 | 3.18 | 2.64 | 4.66 | 4.53 | 36.35 |
| 'I. meda-60' | 2.00 | 1.99 | 0.22 | 1.08 | 1.78 | 1.40 | −1.01 | −0.20 | −1.48 | 5.78 |
| 'I. meda-61' | 1.82 | 0.42 | −1.09 | 0.69 | 0.47 | −0.57 | −1.55 | 0.69 | 0.08 | 0.96 |
| 'I. meda-62' | 1.09 | 0.08 | 1.41 | 0.21 | 0.24 | 1.51 | −0.39 | −1.46 | −1.88 | 0.81 |
| 'I. meda-63' | 1.02 | 0.48 | 0.82 | −1.15 | −1.45 | −1.94 | −0.60 | 0.36 | −0.43 | −2.89 |
| 'I. meda-64' | −0.25 | 1.62 | −0.61 | 0.06 | 1.13 | −0.41 | 0.49 | 1.45 | 1.80 | 5.28 |
| 'I. meda-65' | −1.41 | 1.71 | −0.03 | −0.97 | −0.16 | 1.92 | −0.03 | −0.68 | 0.53 | 0.88 |
| 'I. meda-66' | −1.04 | −1.70 | −1.48 | −1.49 | −1.39 | −1.44 | 0.56 | −1.27 | −0.62 | −9.87 |
| 'I. meda-67' | 1.59 | −0.10 | 0.67 | −1.31 | −1.23 | −1.84 | −1.32 | −0.89 | −1.29 | −5.72 |
| 'I. meda-68' | −1.65 | −1.52 | −0.16 | −1.17 | −0.54 | 0.01 | 0.76 | −1.84 | 1.20 | −4.91 |
| 'I. meda-69' | 0.51 | −1.67 | 1.49 | 1.68 | −1.76 | −0.89 | 1.22 | 0.99 | −1.26 | 0.31 |
| 'I. meda-70' | −1.16 | −0.52 | −0.06 | 0.47 | −0.52 | −0.15 | 0.99 | −1.85 | −0.99 | −3.79 |
| 'I. meda-71' | 0.85 | 1.58 | 0.05 | 0.13 | −1.57 | −0.21 | 0.13 | −1.03 | −0.92 | −0.99 |
| 'I. meda-72' | −0.49 | −1.92 | −0.71 | −1.15 | −0.69 | −1.52 | 1.56 | 0.37 | 0.72 | −3.83 |
| 'I. meda-73' | 1.16 | −0.01 | −1.65 | 0.15 | 0.35 | 0.98 | −0.27 | −1.49 | −0.86 | −1.64 |
| 'I. meda-74' | −0.55 | 0.58 | 0.28 | −0.58 | 1.95 | 0.42 | −1.05 | −1.59 | −1.39 | −1.93 |
| 'I. meda-75' | −1.02 | −1.36 | −1.25 | −0.86 | −1.31 | 1.59 | −1.68 | 0.10 | −0.36 | −6.15 |
| 'I. meda-76' | 1.93 | −1.55 | −0.41 | 1.88 | 1.46 | 1.27 | −0.97 | −1.32 | 0.67 | 2.96 |
| 'I. meda-77' | 1.72 | 0.23 | 0.29 | −0.88 | 1.08 | −1.25 | −0.71 | −0.30 | 0.03 | 0.21 |
| 'I. meda-78' | −1.03 | −1.54 | 0.44 | −0.85 | 0.32 | −1.38 | −0.08 | 0.13 | −1.79 | −5.78 |
| 'I. meda-79' | −0.65 | −1.46 | −1.75 | 1.96 | −0.71 | 1.24 | −0.98 | 0.73 | 1.04 | −0.58 |
| 'I. meda-80' | 0.38 | −0.11 | −0.35 | −0.60 | 1.72 | 1.32 | 1.86 | −1.50 | 0.92 | 3.64 |
| 'I. meda-81' | 4.85 | 2.95 | 2.67 | 4.35 | 3.94 | 4.60 | 2.85 | 4.49 | 3.00 | 33.70 |
| 'I. meda-82' | 2.91 | 2.91 | 4.54 | 4.16 | 3.81 | 3.40 | 4.69 | 3.48 | 4.54 | 34.44 |
| 'I. meda-83' | −0.24 | −0.49 | −0.15 | −0.79 | 0.99 | 0.01 | −1.07 | 1.60 | −0.46 | −0.60 |
| 'I. meda-84' | 0.17 | 1.63 | 0.50 | −1.53 | 1.76 | 0.51 | −0.66 | −1.44 | 1.18 | 2.12 |
| 'I. meda-85' | 4.05 | 3.83 | 4.73 | 4.47 | 2.88 | 3.28 | 3.12 | 4.36 | 2.58 | 33.30 |
| 'I. meda-86' | 0.28 | 1.05 | 1.51 | −0.63 | 1.29 | −1.56 | 1.39 | −1.49 | −0.41 | 1.43 |
| 'I. meda-87' | 1.19 | −1.40 | −1.08 | 0.89 | 0.88 | 0.56 | 0.78 | 0.17 | −0.99 | 1.00 |
| 'I. meda-88' | −0.62 | −1.27 | 1.63 | 0.33 | −0.40 | −0.15 | 1.79 | −1.39 | 0.34 | 0.26 |
| 'I. meda-89' | 0.02 | 0.45 | −1.93 | 1.49 | 1.73 | 0.26 | 0.79 | 1.69 | 0.83 | 5.33 |
| 'I. meda-90' | −1.39 | 0.31 | 0.43 | −0.30 | 0.95 | 1.74 | 1.70 | −0.20 | −1.55 | 1.69 |
| 'I. meda-91' | 1.94 | 1.36 | −1.50 | 1.68 | 1.48 | 0.08 | 0.37 | −0.40 | −1.78 | 3.23 |

*(Continued)*

Table 5. (Continued)

| Accession | PC1 | PC2 | PC3 | PC4 | PC5 | PC6 | PC7 | PC8 | PC9 | Composite score |
|---|---|---|---|---|---|---|---|---|---|---|
| '*I. meda*-92' | −0.66 | 1.21 | −1.98 | −0.67 | −0.41 | 0.15 | 1.68 | −0.61 | −0.61 | −1.90 |
| '*I. meda*-93' | 4.34 | 3.63 | 3.06 | 3.63 | 2.85 | 2.94 | 3.75 | 3.55 | 4.79 | 32.54 |
| '*I. meda*-94' | −0.55 | 0.32 | 0.53 | −1.95 | 0.65 | −1.29 | 1.84 | −1.41 | −0.34 | −2.20 |
| '*I. meda*-95' | 2.71 | 4.99 | 3.76 | 3.99 | 2.67 | 4.37 | 3.02 | 4.75 | 3.01 | 33.27 |
| '*I. meda*-96' | −1.24 | −1.85 | −0.11 | 0.26 | −1.74 | 1.10 | −0.19 | 0.10 | −0.24 | −3.91 |
| '*I. meda*-97' | −0.40 | 0.24 | −1.38 | −1.27 | 1.45 | 1.78 | −0.51 | −0.92 | 0.58 | −0.43 |
| '*I. meda*-98' | −0.37 | −1.90 | −1.38 | 0.86 | 0.64 | −1.89 | −1.11 | −1.08 | 0.69 | −5.54 |
| '*I. meda*-99' | 2.55 | 2.76 | 4.50 | 2.95 | 4.13 | 3.10 | 2.75 | 3.11 | 4.31 | 30.16 |
| '*I. meda*-100' | 1.42 | 1.32 | −0.41 | 0.67 | −1.18 | −0.83 | 1.59 | −1.95 | −1.66 | −1.03 |
| '*I. meda*-101' | −1.17 | −1.89 | −1.27 | 0.33 | −0.31 | 1.57 | 1.27 | −0.63 | −0.96 | −3.06 |
| '*I. meda*-102' | −0.48 | 0.36 | −0.93 | 0.50 | −0.36 | 0.21 | −0.26 | −0.82 | 1.79 | 0.01 |
| '*I. meda*-103' | 1.05 | −1.44 | 1.47 | −0.05 | 1.58 | 1.20 | −0.30 | −1.91 | −0.93 | 0.67 |
| '*I. meda*-104' | 0.17 | 0.53 | −0.97 | −1.44 | 1.34 | 1.94 | 0.10 | −1.31 | −0.91 | −0.55 |
| '*I. meda*-105' | −1.93 | 1.66 | −1.53 | 0.31 | −0.90 | 0.22 | 0.61 | 1.32 | −1.17 | −1.41 |
| '*I. meda*-106' | −1.96 | −1.45 | 1.60 | 1.50 | 0.39 | 0.40 | 0.66 | −1.30 | 1.66 | 1.50 |
| '*I. meda*-107' | 3.55 | 3.46 | 3.80 | 2.62 | 2.92 | 4.35 | 2.71 | 4.01 | 3.11 | 30.53 |
| '*I. meda*-108' | −0.44 | −0.85 | −0.58 | 0.88 | −0.81 | 0.27 | −0.10 | 0.65 | 1.75 | 0.77 |

process, derived using Euclidean distance metrics. Together, these results offer a thorough perspective on the morphological divergence and potential genetic structuring among the 108 wild accessions collected from the Tafresh region. The dendrogram displays a well-defined hierarchical arrangement of morphological similarities, where accessions are sequentially merged into broader clusters with increasing linkage distance. This hierarchical structure reflects varying degrees of phenotypic differentiation, with closely grouped individuals exhibiting greater morphological resemblance.

At the initial clustering step, for example, accessions '*I. meda*-23' and '*I. meda*-95' were joined at a minimal distance of 4.43, indicative of strong morphological similarity. As the clustering progressed, fusion distances gradually increased, culminating at 47.29 in the final stage—where accessions 1 and 11, representing the most morphologically distinct genotypes, were ultimately grouped. The broad range of fusion distances (4.43–47.29) points to substantial morphological heterogeneity within the population. Notably, substantial increases in linkage distance at later clustering stages (e.g., stages 106 and 107) suggest the joining of well-separated clusters, potentially corresponding to distinct morphotypes or subpopulations with underlying genetic differentiation.

Accessions exhibiting such pronounced divergence merit further ecological or molecular investigation, as they may represent rare genotypes, unique adaptive traits, or relic lineages. For instance, accessions '*I. songarica*-11' and '*I. songarica*-33' were merged during the later clustering stages, underscoring their marked phenotypic distinctiveness. The final linkage of '*I. songarica*-1' and '*I. songarica*-11' at the highest Euclidean distance further reinforces the potential uniqueness of these accessions within the dataset.

Overall, the hierarchical clustering results confirm the existence of extensive intraspecific morphological variation within *I. meda*, structured into both closely related clusters and highly divergent individuals. This phenotypic organization suggests a rich reservoir of genetic diversity within the Tafresh population, likely shaped by environmental variability, geographic separation, and localized selective pressures. These findings carry important implications for taxonomy, ecology, and the development of conservation and breeding programs, particularly in identifying and preserving phenotypically and genetically distinct accessions.

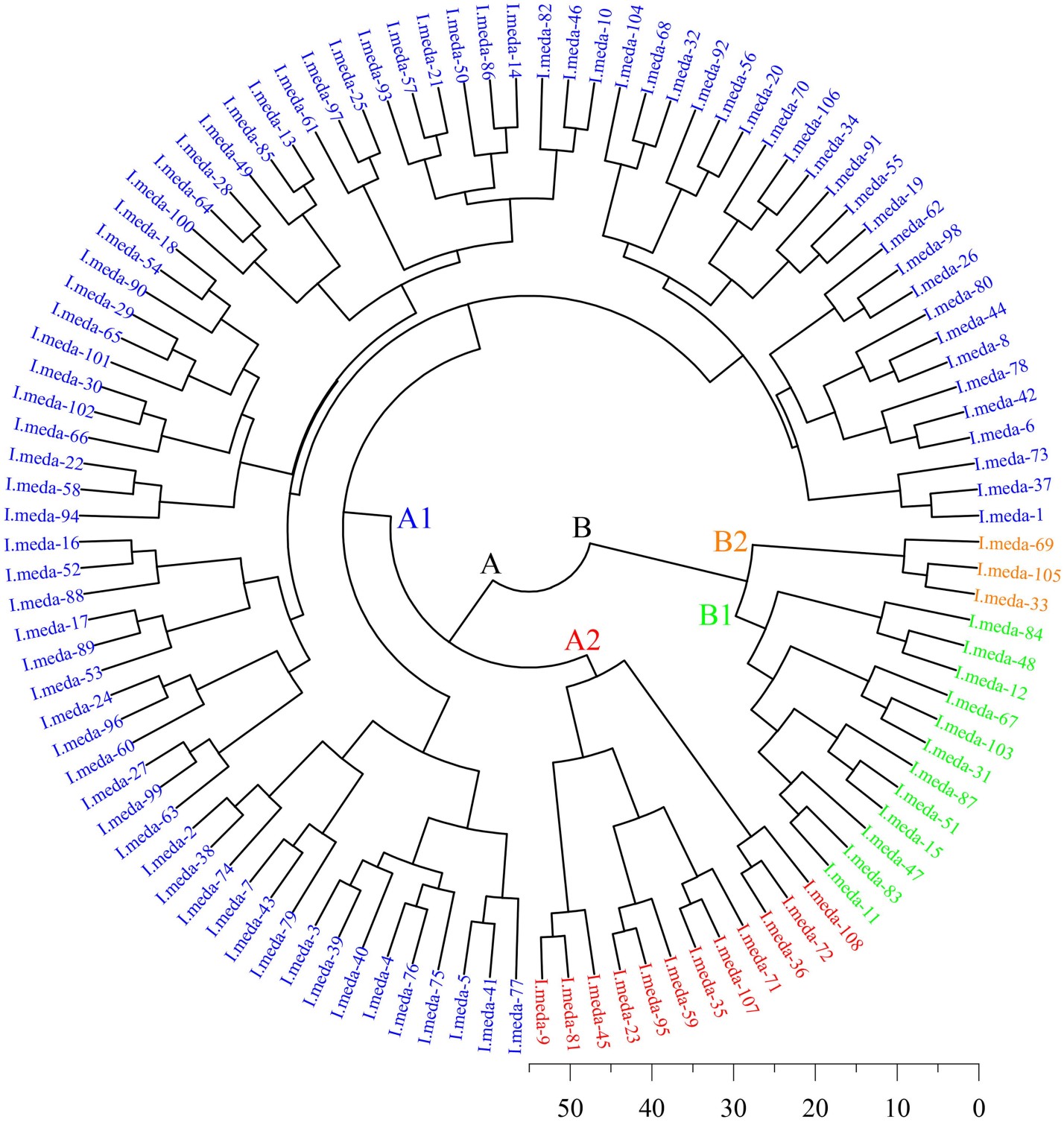

**Fig 7. Hierarchical cluster analysis-based visualization of morphological clustering patterns among *Iris meda* accessions.**

Comparable clustering-based methodologies have been widely adopted in previous studies to explore morphological diversity and genetic structure in *Iris* species. For example, Azimi et al. [31] applied HCA to Iranian *Iris* taxa, identifying discrete phenotypic groups indicative of underlying genetic variability. In a subsequent study, Azimi et al. [32] utilized multivariate analyses, including HCA, to classify *Iris germanica* hybrids, facilitating informed decisions in hybrid selection. Asgari et al. [33] also employed HCA to examine morphological diversity among wild *Iris* species with ornamental value, demonstrating the method's effectiveness in detecting superior phenotypes. Similarly, Ghorbani et al. [34] used cluster analysis to characterize morphological differentiation among *Iris pseudacorus* accessions, highlighting ecotypes with potential conservation significance. The application of PCA and hierarchical clustering for evaluating diversity and grouping accessions has been successfully applied in leafy greens and other crops [47,48].

## 4. Conclusions

This comprehensive morphometric assessment of 108 wild *I. meda* accessions from the Tafresh region of Iran has revealed a striking degree of phenotypic diversity across both vegetative and floral traits. The high coefficients of variation observed in key descriptors—such as leaf shape (62.56%), crest color (59.79%), and standard color (46.35%)—highlight the extensive genetic variability present within the species. Such diversity is likely shaped by ecological heterogeneity, microhabitat differentiation, and potential gene flow across populations, and it underpins the adaptive capacity and ornamental value of *I. meda*.

Multivariate statistical analyses, including PCA, correlation matrix, and multiple regression, identified flower diameter, fall length, and spathe length as the most discriminative traits contributing to morphological divergence. These traits are not only essential for ecological adaptation and pollinator attraction but also represent primary targets for breeding programs. Regression analyses emphasized the developmental coordination between floral dimensions and supporting vegetative features—such as the relationships between beard length and fall length ($\beta = 0.63$), and flower diameter and floral surface ($r = 0.80$; $\beta = 0.87$)—revealing a tightly integrated morphological architecture within the species.

PCA and hierarchical clustering analyses revealed a subset of highly divergent accessions, notably '*I. meda-10*', '*I. meda-23*', '*I. meda-27*', '*I. meda-59*', '*I. meda-81*', '*I. meda-82*', '*I. meda-85*', '*I. meda-93*', '*I. meda-95*', '*I. meda-99*', and '*I. meda-107*', which were located outside the 95% confidence ellipse of the PCA biplot and exhibited the highest PC scores. These genotypes represent morphologically unique individuals with substantial breeding and conservation value. On the other hand, elite accessions with exceptional floral traits—such as '*I. meda-3*' (maximum spathe length: 71.88 mm), '*I. meda-7*' (maximum fall length: 54.63 mm), and '*I. meda-36*' (maximum flower diameter: 71.23 mm)—were identified as promising candidates for ornamental enhancement and horticultural exploitation.

In conclusion, this study not only provides a detailed phenotypic profile of *I. meda* but also identifies morphologically outstanding accessions that can serve as valuable genetic resources. The accessions '*I. meda-3*', '*I. meda-7*', and '*I. meda-36*' are particularly recommended for ornamental enhancement, while the genetically divergent genotypes '*I. meda-10*', '*I. meda-23*', '*I. meda-27*', '*I. meda-59*', '*I. meda-81*', '*I. meda-82*', '*I. meda-85*', '*I. meda-93*', '*I. meda-95*', '*I. meda-99*', and '*I. meda-107*' offer promising material for conservation and breeding strategies. These findings reinforce the utility of integrative morphometric approaches in uncovering intra-specific diversity and guiding effective germplasm utilization in *Iris* species.

The real necessity of this study lies in establishing a foundational morphological dataset for *I. meda*, a species with limited prior characterization despite its ecological and ornamental significance. By documenting its phenotypic richness, identifying elite and divergent accessions, and recommending the integration of molecular markers (e.g., SSRs, SNPs, or NGS-based approaches), this work provides indispensable baseline information for breeding, conservation, and sustainable utilization. Combining morphometric data with molecular and ecological analyses will further validate the observed variation, clarify genetic structure, strengthen breeding programs, facilitate the identification of adaptive traits under climate change scenarios, and enhance conservation strategies for this narrowly distributed yet genetically rich species.

## Supporting information

**S1 Table. Cluster formation stages of _Iris meda_ accessions.**
(DOCX)

**S1 File. Inclusivity-in-global-research-questionnaire.**
(DOCX)

## Acknowledgments

None.

## Author contributions

**Conceptualization:** Alireza Khaleghi, Yazgan Tunç.

**Data curation:** Ali Khadivi, Yazgan Tunç.

**Formal analysis:** Alireza Khaleghi, Ali Khadivi, Yazgan Tunç.

**Investigation:** Alireza Khaleghi, Ali Khadivi.

**Methodology:** Alireza Khaleghi, Ali Khadivi, Yazgan Tunç.

**Validation:** Ali Khadivi.

**Writing – original draft:** Alireza Khaleghi, Ali Khadivi, Yazgan Tunç.

**Writing – review & editing:** Alireza Khaleghi, Ali Khadivi, Yazgan Tunç.

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
