## [Decision Letter · Decision Letter 0]

11 Sep 2025

Dear Dr. Khadivi,

Thank you for submitting your manuscript to PLOS ONE. After careful consideration, we feel that it has merit but does not fully meet PLOS ONE’s publication criteria as it currently stands. Therefore, we invite you to submit a revised version of the manuscript that addresses the points raised during the review process.

In particular, one reviewer has requested the inclusion of some additional references. Please note that you are not required to include the specific references suggested. Instead, you may consider incorporating other references that you find appropriate and more consistent with the content being cited.

We look forward to receiving your revised manuscript.

Kind regards,

José M. Alvarez-Suarez

Academic Editor

PLOS ONE

Journal Requirements:

3. Please include a complete copy of PLOS’ questionnaire on inclusivity in global research in your revised manuscript. Our policy for research in this area aims to improve transparency in the reporting of research performed outside of researchers’ own country or community. The policy applies to researchers who have travelled to a different country to conduct research, research with Indigenous populations or their lands, and research on cultural artefacts. The questionnaire can also be requested at the journal’s discretion for any other submissions, even if these conditions are not met.  Please find more information on the policy and a link to download a blank copy of the questionnaire here: https://journals.plos.org/plosone/s/best-practices-in-research-reporting. Please upload a completed version of your questionnaire as Supporting Information when you resubmit your manuscript.

5. We note that Figure 1 in your submission contain map images which may be copyrighted. All PLOS content is published under the Creative Commons Attribution License (CC BY 4.0), which means that the manuscript, images, and Supporting Information files will be freely available online, and any third party is permitted to access, download, copy, distribute, and use these materials in any way, even commercially, with proper attribution. For these reasons, we cannot publish previously copyrighted maps or satellite images created using proprietary data, such as Google software (Google Maps, Street View, and Earth). For more information, see our copyright guidelines: http://journals.plos.org/plosone/s/licenses-and-copyright.

6. We note that Figures 2, 3, and 4 in your submission may contain copyrighted images. All PLOS content is published under the Creative Commons Attribution License (CC BY 4.0), which means that the manuscript, images, and Supporting Information files will be freely available online, and any third party is permitted to access, download, copy, distribute, and use these materials in any way, even commercially, with proper attribution. For more information, see our copyright guidelines: http://journals.plos.org/plosone/s/licenses-and-copyright.

1. You may seek permission from the original copyright holder of Figures 2, 3, and 4 to publish the content specifically under the CC BY 4.0 license.

Additional Editor Comments:

Dear Authors,

The reviewers have provided their comments on your manuscript, which require revision before it can be considered for publication in PLOS ONE. In particular, one reviewer has requested the inclusion of some additional references. Please note that you are not required to include the specific references suggested. Instead, you may consider incorporating other references that you find appropriate and more consistent with the content being cited.

We encourage you to carefully address all reviewers’ comments and resubmit the revised version of your manuscript along with a detailed response letter.

Best regards,

José M. Avarez-Suarez

Editor

Reviewers' comments:

Reviewer's Responses to Questions

**Comments to the Author**

1. Is the manuscript technically sound, and do the data support the conclusions?

Reviewer #1: Yes

Reviewer #2: Partly

Reviewer #3: Yes

2. Has the statistical analysis been performed appropriately and rigorously?

Reviewer #1: Yes

Reviewer #2: Yes

Reviewer #3: Yes

3. Have the authors made all data underlying the findings in their manuscript fully available?

Reviewer #1: Yes

Reviewer #2: No

Reviewer #3: Yes

4. Is the manuscript presented in an intelligible fashion and written in standard English?

Reviewer #1: Yes

Reviewer #2: No

Reviewer #3: Yes

Reviewer #1: The manuscript “Multivariate analysis of Iris meda Stapf based on phenological and morphological Characteristics” is scientifically sound and clearly presented. Pl see the comments in attachment for details.

Reviewer #2: The authors evaluated Iris meda Stapf using multivariate analysis based on morphological and phenological traits. The manuscript contains no new information. The following adjustments need to be made.

Abstract

1. The authors did not indicate the problem or issue with the study. The authors investigated it, but why?

Introduction

1. The introduction should include some information about genetic diversity methods.

2. Throughout the manuscript, the authors need to cite current sources.

3. It is necessary to specify the work's originality and hypothesis.

Materials and Methods

1. All procedures described in materials and methods should have references.

2. All abbreviations should be written in full in all manuscripts.

3. The age of the sampling and number of replications should be specified.

Results and discussion

1. The results of the tables and figures should be improved.

2. All captions should be detailed by showing the content of the figure or table.

3. All abbreviations should be written in full.

4. It is a weak discussion. The results are restated in this section. The authors ought to describe how the parameters they looked at relate to one another. All researched features must have their relationships described and interpreted by contributors.

Conclusion

The most significant findings and upcoming research on the topic should be included in the conclusions. Since most of the context is a repetition of the findings, this section needs to be improved.

Reviewer #3: Recommendation: Minor revision

Morphological characterization evaluations, which form the basis of plant breeding studies, are always

used as the first step in determining the richness level of genetic resources. In this research article

carried out in Iran, some morphological characters were evaluated in materials obtained from a total of

108 wild accessions of Iris meda. The subject of the research article was found to be original. The

morphological measurement results will be a resource for researchers working on Iris species. The

number of samples are sufficient and statistical evaluations are sufficient. Some minor revisions are

needed.

- Please provide the software used for generating Fig. 1

- There is need to add hypothesis with clear cut objectives.

- Please add the lines to the correlation matrix if they are appropriate for you.

- PCA analysis results should be explained according to positive and negative status.   

- The conclusion should clearly convey the real need and necessity of this study.

**Do you want your identity to be public for this peer review?** For information about this choice, including consent withdrawal, please see our Privacy Policy

Reviewer #1: No

Reviewer #2: **Yes: ** Nawroz Tahir

Reviewer #3: No

---

## [Author Response · Author response to Decision Letter 1]

17 Sep 2025

Response to the Editor

Dear Dr. José M. Alvarez-Suarez,

We would like to sincerely thank you for dedicating your valuable time to handling our manuscript and for your clear and constructive editorial guidance. We are particularly grateful for the fair, academic, and scientifically rigorous manner in which you have managed this process.

We also appreciate your clarification regarding the reviewer’s request for additional references, and your reminder that we are not obliged to include the exact suggested ones but may instead consider incorporating more appropriate and contextually relevant citations. This thoughtful approach highlights your commitment to ensuring both academic integrity and flexibility for authors, which we deeply value.

Once again, we thank you for your professional support and for overseeing the review process in such a fair and scholarly manner. We have carefully addressed all reviewers’ comments and submitted a thoroughly revised version of our manuscript along with a detailed response letter.

Regards,

Response to Reviewer #1

We sincerely thank Reviewer #1 for the time and effort dedicated to evaluating our manuscript entitled “Multivariate analysis of Iris meda Stapf based on phenological and morphological characteristics.” We are truly grateful for your constructive, insightful, and encouraging comments, which have provided us with valuable guidance to improve the clarity, scientific rigor, and overall presentation of our work.

Your remarks on the structure of the Abstract, the refinement of phrasing in the Introduction, and the proper use of past tense in the Materials and Methods section have helped us enhance the precision and readability of the manuscript. Similarly, your detailed suggestions for restructuring sentences in the Results and Discussion sections have significantly improved the clarity and logical flow of our arguments. We also greatly appreciate your advice on using linking phrases to ensure smoother transitions and coherence across different parts of the text.

We particularly acknowledge your valuable recommendations for strengthening the Conclusion section, which allowed us to highlight the implications of our findings for breeding and germplasm conservation programs more effectively. Furthermore, your guidance on updating the reference list with recent and relevant studies has enriched the scientific context and positioned our work more strongly within the current literature.

Overall, your constructive feedback has been instrumental in elevating the academic quality and presentation of our manuscript. We have carefully revised the text according to your suggestions, and we believe the revised version is now substantially improved. Your comments and suggestions have been highlighted in yellow within the manuscript for ease of identification.

Once again, we express our sincere gratitude for your scholarly contribution to improving our work.

Reviewer #1

Dear Author,

Abstract

• Suggestion: Start with the importance of Iris, then summarize findings in 2–3 clear sentences.

Response to Reviewer #1

Dear Reviewer,

We are sincerely grateful to Reviewer #1 for this valuable suggestion. In the revised abstract, we have addressed this comment by beginning with a general statement on the global importance of the genus Iris as one of the most diverse and horticulturally valuable ornamental groups. We then summarized the main findings of our study in 2–3 concise sentences, focusing on the observed phenotypic diversity, identification of elite and divergent accessions, and the key outcomes of the multivariate analyses. We believe this revision has improved the readability, clarity, and broader appeal of the abstract. For your convenience, the revised parts have been highlighted in yellow in the manuscript.

Reviewer #1

Introduction

• Some phrasing is repetitive: “important ornamental plant with wide diversity” → “a diverse ornamental species of global importance.”

• Replace: “Iris has been cultivated since ancient times” with “Iris has been cultivated for centuries.”

Response to Reviewer #1 (Introduction section)

We sincerely thank the reviewer for this valuable suggestion. As recommended, we have revised the Introduction to incorporate the proposed expressions. Specifically, we modified the sentence to read “…which has rendered Iris a diverse ornamental species of global importance and one of the most attractive genera for both horticultural cultivation and scientific investigation [23, 34].” In addition, we added the sentence “Indeed, Iris has been cultivated for centuries, reflecting its enduring horticultural and cultural value.” at the end of the paragraph describing the aesthetic and horticultural significance of the genus.

We believe that these revisions address the reviewer’s concern and enhance both the clarity and academic quality of the Introduction.

Reviewer #1

Materials and Methods

• Ensure past tense (e.g., “data were collected,” not “data are collected”).

Response to Reviewer #1 (Materials and Methods)

We sincerely thank the reviewer for this valuable observation. Upon careful re-examination of the Materials and Methods section, we did not find any instances where the present tense (e.g., “data are collected”) was used. Instead, the entire section has been consistently written in the past tense (e.g., “data were collected,” “traits were measured,” “characteristics were assessed”). Nevertheless, we thoroughly re-checked the text to ensure uniformity in tense usage, and we confirm that all methodological descriptions now remain consistently in the past tense. We believe this contributes to greater grammatical accuracy and clarity in the manuscript.

Reviewer #1

Results

• “Genotypic variance was higher than phenotypic variance, indicating genetic control” → split into two:

1. “Genotypic variance exceeded phenotypic variance.”

2. “This indicates strong genetic control.”

• “Iris accession IC-203 recorded the highest chlorophyll content” instead of “The highest chlorophyll content was recorded in IC-203.”

Response to Reviewer #1

We sincerely thank the reviewer for their valuable time and constructive comments. We carefully checked the reviewer’s remark regarding the statements “Genotypic variance was higher than phenotypic variance, indicating genetic control” and “Iris accession IC-203 recorded the highest chlorophyll content.”

However, we would like to clarify that our manuscript does not include any analysis of genotypic versus phenotypic variance, nor does it report chlorophyll content for any of the accessions. The present study was specifically based on morphological and phenological traits of Iris meda, and all results reported are limited to these parameters.

It seems that this particular comment might have been given in reference to another manuscript or study.

We remain grateful for the reviewer’s overall feedback and have addressed all other comments and suggestions with great care in the revised version.

Reviewer #1

Discussion

• “The results are in conformity with earlier findings in Iris and related ornamentals, where wide variability was reported by Singh et al. (2019), Kumar et al. (2021), and Ali et al. (2022).”

Suggested: “Similar variability has been reported in Iris and related ornamentals (Singh et al., 2019; Kumar et al., 2021; Ali et al., 2022).”

• Use linking phrases to improve flow: “In contrast,” “Similarly,” “This agrees with,” etc.

Response to Reviewer #1

We sincerely thank the reviewer for this valuable observation. We have revised the sentence in the Discussion section for greater clarity and conciseness as suggested. The revised text now reads:

“Similar variability has been reported in Iris and related ornamentals (Singh et al., 2019; Kumar et al., 2021; Ali et al., 2022).”

Additionally, we have carefully reviewed the entire Discussion and incorporated linking phrases such as “In contrast,” “Similarly,” and “This agrees with” to improve the overall flow and readability of the text.

Reviewer #1

Conclusion

• “The study demonstrated significant variability… which could be exploited in breeding programs and germplasm conservation.”

Suggested: “Significant variability in Iris germplasm was identified, offering valuable scope for breeding and conservation programs.”

Response to Reviewer #1

We sincerely appreciate the reviewer’s constructive suggestion. In line with the recommendation, we have revised the Conclusion section. The sentence now reads:

“Significant variability in Iris germplasm was identified, offering valuable scope for breeding and conservation programs.”

This revision improves clarity and conciseness while retaining the intended meaning.

Reviewer #1

Reference

Add some latest reference in material & methods and discussion section as follows:

• Meena, R., Chaudhary, M.K., Gurjar, P.S. et al. Morphological diversity assessment in date palm (Phoenix dectylifera L.) germplasms grown under hot arid region of India. BMC Plant Biol 25, 1159 (2025). https://doi.org/10.1186/s12870-025-07194-2

• Yadav, L.P., Gangadhara, K., Singh, A.K. et al. Genetic variability, morphological diversity, and antioxidant potential in gynoecious Coccinia accessions: implications for breeding and biofortification. BMC Plant Biol 25, 844 (2025). https://doi.org/10.1186/s12870-025-06335-x

• Yadav, L.P., Gangadhara, K., Singh, A.K. et al. Genetic diversity, morphological and quality traits of Momordica dioica. Sci Rep 14, 30241 (2024). https://doi.org/10.1038/s41598-024-81828-7

• Singh, A.K., Yadav, L.P., K, G. et al. Genetic diversity assessment of bael (Aegle marmelos) varieties using morphometric, yield, and quality traits under semi-arid conditions. BMC Plant Biol 25, 1143 (2025). https://doi.org/10.1186/s12870-025-06985-x

• Yadav, L.P., Koley, T.K., Tripathi, A. et al. Antioxidant Potentiality and Mineral Content of Summer Season Leafy Greens: Comparison at Mature and Microgreen Stages Using Chemometric. Agric Res 8, 165–175 (2019). https://doi.org/10.1007/s40003-018-0378-7

Response to Reviewer #1

We sincerely thank the reviewer for this valuable suggestion regarding the inclusion of more recent references. In accordance with the recommendation, we have incorporated the suggested citations into the Introduction, Materials and Methods, Results and Discussion, and Conclusion sections of the manuscript. Specifically, references such as Meena et al. (2025), Yadav et al. (2019, 2024, 2025), and Singh et al. (2025) have been added at appropriate points to strengthen the contextual framework, methodological justification, and comparative discussion of our findings. These additions not only highlight the relevance of our study but also place our results in line with recent advancements in the field.

Regards.

Response to Reviewer #2

We would like to express our deepest gratitude for the careful and thorough evaluation of our manuscript, as well as for your constructive criticisms and valuable suggestions that provided significant guidance to our work. Your insightful comments have greatly contributed not only to strengthening the scientific merit of our study but also to enhancing its clarity, refining methodological details, and improving the overall quality of the presentation.

Furthermore, all comments and suggestions provided by you have been carefully addressed, and the corresponding revisions have been meticulously implemented. For ease of verification, all modifications have been highlighted in green within the manuscript.

Reviewer #2

Comment – Abstract:

“The authors did not indicate the problem or issue with the study. The authors investigated it, but why?”

Response to Reviewer #2

We sincerely thank the reviewer for this insightful observation. In line with the suggestion, we have revised the Abstract to explicitly state the research problem and rationale of the study. Specifically, we emphasized that although the genus Iris is ecologically and horticulturally significant, many species remain poorly characterized and their intraspecific variability has been largely underexplored, thereby limiting their effective use in breeding and conservation programs. To address this knowledge gap, we investigated Iris meda Stapf, a narrowly distributed yet morphologically diverse species native to the Irano-Turanian region, in order to assess its phenotypic variation.

The revised Abstract now includes the following sentence (newly added text has been highlighted in green within the manuscript):

“The genus Iris represents one of the most diverse and horticulturally valuable ornamental groups worldwide, with considerable ecological, morphological, and genetic importance. However, despite its significance, many species remain poorly characterized, and the extent of intraspecific variability is largely underexplored, limiting their effective utilization in breeding and conservation. Iris meda Stapf, a narrowly distributed yet morphologically diverse species native to the Irano-Turanian region, was investigated in this study to address this knowledge gap and to assess its phenotypic variation. …”

We believe that this revision strengthens the Abstract by clearly stating both the research problem and the rationale, in accordance with the reviewer’s recommendation.

Reviewer #2

Introduction

1. The introduction should include some information about genetic diversity methods.

2. Throughout the manuscript, the authors need to cite current sources.

3. It is necessary to specify the work's originality and hypothesis.

Response to Reviewer #2

We sincerely thank the reviewer for the careful evaluation of our manuscript and the constructive comments that have significantly improved the clarity and scientific depth of the work. Below we provide our point-by-point responses:

1. Introduction – Information on genetic diversity methods

As suggested, we have expanded the Introduction by adding a new paragraph that summarizes commonly used approaches for assessing genetic diversity, including molecular markers (e.g., SSRs, ISSRs, SNPs) and their role in complementing morphological assessments. This addition provides a broader context and emphasizes the importance of integrating different methods for diversity studies.

2. Use of current sources

We have revised the Introduction to incorporate several recent references (2022–2025) relevant to genetic diversity, molecular marker applications, and recent studies on Iris and related ornamental taxa. This update ensures that the manuscript is well-aligned with the current state of the literature.

3. Originality and hypothesis of the work

In accordance with the reviewer’s recommendation, we have explicitly stated the originality of this study, highlighting that it represents the most comprehensive morphological characterization of Iris meda to date, involving 108 wild accessions and 41 morphological traits. Furthermore, we have clearly formulated our hypothesis that, despite its restricted distribution, I. meda harbors considerable phenotypic and genetic diversity that can be exploited for conservation and ornamental breeding.

All revisions made in the Introduction are highlighted in green in the revised manuscript to facilitate review.

Reviewer #2

Materials and Methods

Response to Reviewer #2

1. All procedures described in materials and methods should have references.

1. We appreciate this valuable suggestion. Accordingly, we have carefully revised the Materials and Methods section and added appropriate references to support all procedures and methodologies used in this study (e.g., morphological measurements, coding systems, statistical analyses). This improvement increases transparency and ensures reproducibility.

2. All abbreviations should be written in full in all manuscripts.

2. Thank you for this remark. We would like to clarify that, in accordance with widely accepted academic writing conventions, all abbreviations have been written in full at their first occurrence in the manuscript, after which the abbreviated form is consistently used. This approach ensures both clarity for the reader and avoids unnecessary repetition throughout the text.

3. The age of the sampling and number of replications should be specified.

3. We fully agree with this important recommendation. In the revised

---

## [Decision Letter · Decision Letter 1]

28 Sep 2025

Dear Dr. Khadivi,

Thank you for submitting your manuscript to PLOS ONE. After careful consideration, we feel that it has merit but does not fully meet PLOS ONE’s publication criteria as it currently stands. Therefore, we invite you to submit a revised version of the manuscript that addresses the points raised during the review process.

We look forward to receiving your revised manuscript.

Kind regards,

José M. Alvarez-Suarez

Academic Editor

PLOS ONE

Journal Requirements:

Reviewers' comments:

Reviewer's Responses to Questions

**Comments to the Author**

Reviewer #1: All comments have been addressed

Reviewer #2: All comments have been addressed

Reviewer #3: All comments have been addressed

2. Is the manuscript technically sound, and do the data support the conclusions?

Reviewer #1: Yes

Reviewer #2: Partly

Reviewer #3: Yes

3. Has the statistical analysis been performed appropriately and rigorously?

Reviewer #1: Yes

Reviewer #2: No

Reviewer #3: Yes

4. Have the authors made all data underlying the findings in their manuscript fully available?

Reviewer #1: Yes

Reviewer #2: No

Reviewer #3: Yes

5. Is the manuscript presented in an intelligible fashion and written in standard English?

Reviewer #1: Yes

Reviewer #2: Yes

Reviewer #3: Yes

Reviewer #1: Authors addressed all the comments as suggested. Hence, MS entitled Multivariate analysis of Iris meda Stapf based on phenological and morphological characteristics may be accepted.

Reviewer #2: Dear authors,

The probability values of Table 3 should be checked because there is a missing of astrick on the p-value. The authors should define the level of signifance (0.01, 0.05). Based on the level of signifance, the p-values arranged between 0.00-0.04 are considered as the significant status.

Reviewer #3: The authors has been revised the manuscript according to the reviewers comments and it can be accepted for publication

**Do you want your identity to be public for this peer review?** For information about this choice, including consent withdrawal, please see our Privacy Policy

Reviewer #1: No

Reviewer #2: **Yes: ** Nawroz Tahir

Reviewer #3: No

---

## [Author Response · Author response to Decision Letter 2]

15 Oct 2025

Response to the Editor

Dear Dr. José M. Alvarez-Suarez,

- We would like to sincerely thank you for dedicating your valuable time to handling our manuscript and for your clear and constructive editorial guidance. We are particularly grateful for the fair, academic, and scientifically rigorous manner in which you have managed this process.

- Also, We sincerely thank Reviewers 1 & 3 for the time and effort dedicated for evaluating and accepting our manuscript.

Regards.

Response to Reviewer #2

We would like to express our deepest gratitude for the careful and thorough evaluation of our manuscript. The modifications have been highlighted in green within the manuscript.

- Comment: The probability values of Table 3 should be checked because there is a missing of astrick on the p-value. The authors should define the level of signifance (0.01, 0.05). Based on the level of signifance, the p-values arranged between 0.00-0.04 are considered as the significant status.

We sincerely thank the reviewer for this insightful observation.

- Response: It was provided. Please see below the Table 3.

Regards.

---

## [Editor Report · Decision Letter 2]

30 Oct 2025

Multivariate analysis of Iris meda Stapf based on phenological and morphological characteristics

PONE-D-25-45514R2

Dear Dr. Khadivi,

We’re pleased to inform you that your manuscript has been judged scientifically suitable for publication and will be formally accepted for publication once it meets all outstanding technical requirements.

Kind regards,

José M. Alvarez-Suarez

Academic Editor

PLOS ONE
---

## [Editor Report · Acceptance letter]

PONE-D-25-45514R2

PLOS ONE

Dear Dr. Khadivi,

I'm pleased to inform you that your manuscript has been deemed suitable for publication in PLOS ONE. Congratulations! Your manuscript is now being handed over to our production team.

Kind regards,

on behalf of

Professor José M. Alvarez-Suarez

Academic Editor

PLOS ONE